# T2I-REASONBENCH: BENCHMARKING REASONING-INFORMED TEXT-TO-IMAGE GENERATION

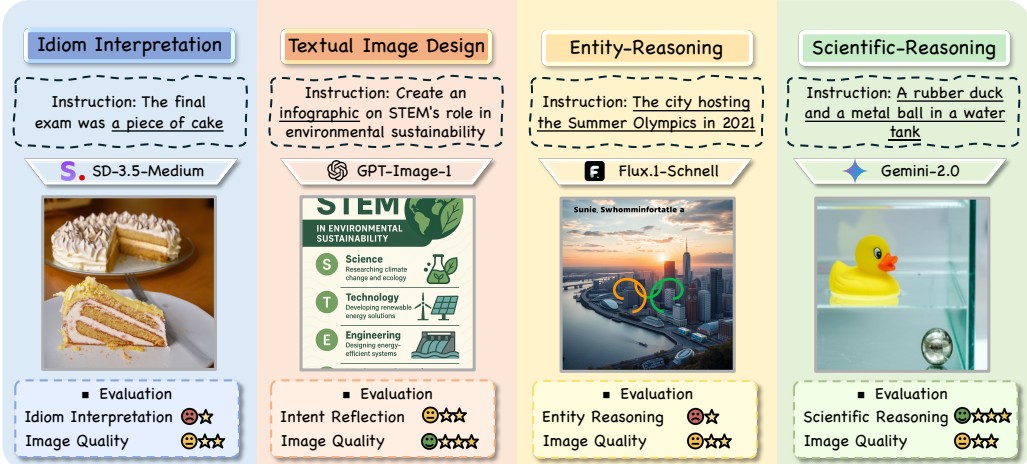

Figure 1: **Overview of T2I-ReasonBench.** We propose T2I-ReasonBench, a benchmark evaluating reasoning capabilities of text-to-image (T2I) models. It consists of four dimensions: **Idiom Interpretation, Textual Image Design, Entity-Reasoning and Scientific-Reasoning**. We propose a two-stage evaluation protocol to measure T2I-ReasonScore, a quantitative metric that integrates reasoning accuracy, detail faithfulness and image quality. We benchmark various T2I generation models, and provide comprehensive analysis on their performances.

## ABSTRACT

Text-to-image (T2I) generative models have achieved remarkable progress, demonstrating exceptional capability in synthesizing high-quality images from textual prompts. While existing research and benchmarks have extensively evaluated the ability of T2I models to follow the literal meaning of prompts, their ability to reason over prompts to uncover implicit meaning and contextual nuances remains underexplored. To bridge this gap, we introduce T2I-ReasonBench, a novel benchmark designed to explore the reasoning capabilities of T2I models. T2I-ReasonBench comprises 800 meticulously designed prompts organized into four dimensions: **(1) Idiom Interpretation**, **(2) Textual Image Design**, **(3) Entity-Reasoning**, and **(4) Scientific-Reasoning**. These dimensions challenge models to infer implicit meaning, integrate domain knowledge, and resolve contextual ambiguities. To quantify the performance, we introduce a two-stage evaluation framework: a large language model (LLM) generates prompt-specific question-criterion pairs that evaluate if the image includes the essential elements resulting from correct reasoning; a multimodal LLM (MLLM) then scores the generated image against these criteria. Experiments across 16 state-of-the-art T2I and unified multimodal models (UMMs) reveal critical limitations in reasoning-informed generation. Our comprehensive analysis indicates that the bottleneck of current models is in reasoning rather than generation. This finding highlights the critical need to enhance reasoning abilities in the next generation of T2I and unified multimodal systems.

# 1 INTRODUCTION

Recent advancements in T2I generative models have enabled the creation of visually appealing images from textual prompts. However, these models often struggle with generating complex scenes that demand reasoning.

Current benchmarks (Yu et al., 2022; Hu et al., 2023; Huang et al., 2023; Ghosh et al., 2023; Hu et al., 2024; Li et al., 2024a; Wu et al., 2024; Huang et al., 5555; Wei et al., 2025), such as T2I-CompBench (Huang et al., 5555) and PartiPrompts (Yu et al., 2022), primarily evaluate literal prompt-image alignment, focusing on object attributes (e.g., color, attribute, count) and relationships. While DPG-Bench (Hu et al., 2024) extends evaluation to long-text comprehension, it remains confined to multi-object composition tasks. These frameworks fail to test models' ability to reason beyond explicit instructions. For instance, generating an image of "A beach ball and a marble in a swimming pool" requires not only object composition but also reasoning about physical laws (e.g., inferring the ball floats while the marble sinks). Such reasoning necessitates understanding related scientific knowledge, such as material density and buoyancy, as well as integrating the reasoning process into T2I generation.

To address this gap, we propose **T2I-ReasonBench**, a novel benchmark designed to systematically evaluate the reasoning ability of T2I models in four dimensions: (1) Idiom Interpretation: Deciphering the implicit meanings of idiomatic expressions with the context to generate appropriate images. (2) Textual Image Design: Understanding the intention of design and effectively planning integrated visual-textual layouts. (3) Entity-Reasoning: applying and integrating the knowledge about world entities in image generation, and (4) Scientific-Reasoning: reasoning with scientific knowledge (e.g., physics, chemistry, biology, astronomy) to produce images adhering to the underlying scientific laws. T2I-ResonBench encompasses the above four dimensions with 800 meticulously designed prompts, all requiring deep reasoning.

To rigorously evaluate performance of T2I models, we introduce a two-stage evaluation framework and propose **T2I-ReasonScore**, a quantitative metric for assessing the quality of reasoning-informed T2I generation. First, an LLM generates specific question-criterion pairs for each prompt. To evaluate the images, an MLLM then answers each question and assigns a score based on the paired criterion. By averaging these scores, we measure how faithfully the image reflects the implicit meaning of the prompt, capturing the effectiveness of model's reasoning. The final T2I-ReasonScore is compose of accuracy and quality perspectives. Our approach allows for fine-grained and interpretable evaluation of models' reasoning ability and addresses the limitation of previous benchmarks that focused solely on literal prompt following.

We evaluate 16 state-of-the-art T2I models, including 8 diffusion models, 5 unified multimodal models, and 3 proprietary models. The results reveal that current models face critical limitations in reasoning-informed generation. Our comprehensive analysis indicates that the bottleneck of current models is in reasoning rather than generation. Although unified multimodal models have better potential in incorporating reasoning, the current models still have a large room for improvement.

Our contributions are threefold: (1) We propose T2I-ReasonBench, a novel benchmark with meticulously designed tasks to explore the reasoning capabilities for text-to-image generation. (2) Our prompt-specific evaluation framework enables fine-grained and interpretable evaluation of reasoning-informed T2I tasks. (3) We evaluate 16 state-of-the-art T2I and unified multimodal models, and provide a thorough analysis of their performances, highlighting notable limitations in reasoning ability of these models.

# 2 RELATED WORK

## 2.1 TEXT-TO-IMAGE GENERATION.

**Diffusion models.** T2I generation has seen rapid advances in recent years, primarily driven by the emergence and refinement of diffusion models (Dhariwal & Nichol, 2021; Ho et al., 2020; Nichol et al., 2021; Saharia et al., 2022). By formulating image synthesis as a progressive denoising process, these models pushed the boundaries of quality and controllability, and established the backbone for modern systems like the Stable Diffusion series (Esser et al., 2024a; Rombach et al., 2022), and the

Flux series (Labs, 2024). Recent models like HiDream (hidream, 2024) and Qwen-Image (Wu et al., 2025) further extend this paradigm, achieving fine-grained, photorealistic text-to-image, solidifying diffusion as the backbone of modern T2I systems.

**Unified multimodal models.** To achieve better token-level alignment between text and image modalities, recent research has shifted toward LLM-based architectures. This includes both autoregressive models, which synthesize images by directly predicting sequences of visual tokens(Ramesh et al., 2021; Ding et al., 2021; Sun et al., 2024; Liu et al., 2024), and unified multimodal models (Team, 2024; Xie et al., 2024; Chen et al., 2025d; Deng et al., 2025; Chen et al., 2025b; Fang et al., 2025; Duan et al., 2025). These unified systems typically combine an autoregressive language model with a diffusion module to integrate understanding and generation. For instance, GoT (Fang et al., 2025) uses an MLLM for semantic-spatial reasoning before diffusion-based synthesis, while Bagel (Deng et al., 2025) unifies an LLM and a diffusion model within a single transformer to generate reasoning chains prior to image creation.

## 2.2 TEXT-TO-IMAGE EVALUATION BENCHMARKS AND METRICS.

**Benchmarks.** Evaluating the capabilities of T2I models requires diverse benchmarks that assess various aspects of understanding and generation. Most of current benchmarks (Yu et al., 2022; Hu et al., 2023; Huang et al., 2023; Ghosh et al., 2023; Hu et al., 2024; Li et al., 2024a; Wu et al., 2024; Huang et al., 5555; Wei et al., 2025) primarily evaluate literal prompt-image alignment. For example, GenEval (Ghosh et al., 2023) utilizes object detection techniques to test whether generated images correctly capture object co-occurrence, position, count, and color described in the prompts. Recent benchmarks have shifted focus from literal alignment to evaluating the reasoning capabilities of T2I models. For instance, Commonsense-T2I (Fu et al., 2024) tests everyday logic through adversarial prompt pairs; PhyBench (Meng et al., 2024) evaluates physical common sense; WISE (Niu et al., 2025) assesses broader world knowledge; and R2I-Bench (Chen et al., 2025c) provides a more comprehensive evaluation across both composition and reasoning categories. These benchmarks collectively establish a multifaceted evaluation landscape, pushing T2I generation towards a deeper understanding of the world.

**Metrics.** Conventional text-image alignment metrics like CLIPscore (Hessel et al., 2021) and VQAscore (Lin et al., 2024) works as bag-of-words models, lacking the expertise needed to evaluate specific composition and reasoning generations. To address this, many works adopt a more targeted, disentangled question-answering framework. Leveraging powerful LLMs and MLLMs, this method first generates specific diagnostic questions and then uses VQA models to answer them by analyzing the image. This approach has been successfully applied across studies evaluating alignment (**?**Cho et al., 2023a; Yarom et al., 2023; Cho et al., 2023b), composition (Wu et al., 2024), reasoning (Chen et al., 2025c), and factual correctness (Lim et al., 2025).

**T2I-ReasonBench vs. Prior Works.** T2I-ReasonBench aims to explore the reasoning abilities of T2I generation through 800 carefully designed prompts across four dimensions. Its key contribution lies in two novel dimensions: Idiom Interpretation and Textual Image Design. Unlike previous benchmarks where the content to generate is well-defined, such as "Einstein's favorite musical instrument" or "A bookshelf with some books, no books on the second shelf", these dimensions assess not only idiom comprehension and text synthesis, but also a model's ability to envision complex scenarios and infer missing information. In Idiom Interpretation, models must generate an image that accurately expresses both a daily scene and the abstract meaning of an idiom. In Textual Image Design, prompts do not clearly specify the text or visual elements needed in the image, therefore models must creatively design and include necessary elements that reflect the prompt's intention, a capability not systematically addressed in prior works.

Furthermore, in Entity-Reasoning dimension, T2I-ReasonBench distinguishes itself by using hard and specific entities that are easily forgotten, rather than common-sense objects, like "A stationery item that removes markings by rubbing against a surface", which refers to the daily item of eraser. This approach more rigorously tests the knowledge integration and reasoning abilities of T2I models.

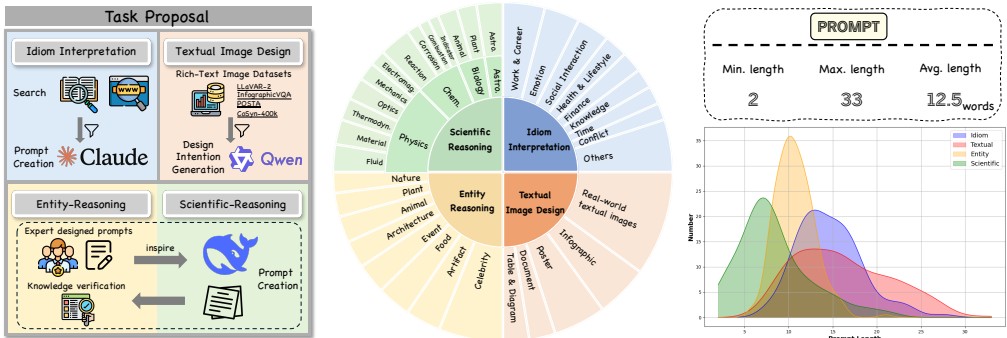

Figure 2: *Left*: Prompt collection process. *Middle*: Subcategories in the four evaluation dimensions. *Right*: Prompt Suite Statistics.

# 3 BENCHMARK CONSTRUCTION

## 3.1 PROBLEM DEFINITION

While modern T2I models are good at explicit prompt-to-image translation, their capacity for reasoning-informed generation remains underexplored. Existing benchmarks focus predominantly on explicit text-image alignment (e.g., object existence, spatial arrangements) but fail to evaluate whether models possess reasoning abilities to uncover the deeper meaning behind the text and generate logically coherent visual content. To this end, we identify four critical scenarios that challenge T2I models to reason about the instructions before visualizing them:

**Scenario 1**: An idiom is a phrase or combination of words with a figurative meaning that differs from its literal meaning. Idioms are common in everyday language, and their meanings usually cannot be deduced by analyzing individual words. For T2I models, prompts containing idioms demand reasoning to obtain the latent meaning before generating semantically faithful visual content. This process requires leveraging linguistic knowledge and effectively analyzing context.

**Scenario 2**: Images with rich text combine visuals and text in various formats. These images are used to serve specific communicative goals, such as education, marketing, and promotion. Generating such content requires T2I models reasoning about the purpose behind the image and apply goal-oriented design skills like layout planning, information structuring, and harmonizing visuals and text.

**Scenario 3**: In everyday life, people often forget specific entity names but remember related details. For example, the prompt "Generate an image of the team lifting the trophy at the 2022 FIFA World Cup" requires the T2I model to reason about the context and then retrieve relevant knowledge to generate the entities not explicitly stated.

**Scenario 4**: Creating physically realistic images remains a significant challenge for current T2I models, which often produce counterintuitive results that violate common sense. This highlights the need to test whether models can apply scientific knowledge. For example, with the prompt "Iron filings scattered around a bar magnet", the model needs to understand magnetism and show the iron filings curving between the magnet's poles.

Based on these four scenarios, we define four dimensions to evaluate the reasoning abilities of T2I models: Idiom Interpretation, Textual Image Design, Entity-Reasoning and Scientific-Reasoning.

## 3.2 PROMPT SUITE OF T2I-REASONBENCH

**Idiom Interpretation.** By sourcing from a book (idi, 2023) and the internet, we collect 200 idioms that are commonly used in daily life but may be challenging for T2I models. We then use an LLM to generate sentences containing the idioms but without explicitly revealing their meanings. These idioms span diverse topics such as social interactions, lifestyle, and emotions. For example, the sentence "He told a funny joke to break the ice at the start of the meeting" uses the idiom "break the ice", which means to ease tension during a first meeting, rather than literally destroying the ice.

**Textual Image Design.** In this category, we first collected 200 images featuring rich text from different datasets. Using an MLLM, we then extracted the underlying design intentions from these images, resulting in 200 design prompts. Each prompt focuses on the functional purpose of the image rather than describing visual details. For example, the prompt "Create a minimalist promotional poster for a workshop on simplicity in design" is an abstract, high-level design instruction. Based on the image sources, the prompts span categories like infographics, posters, documents, tables, diagrams, and other real-world images such as book covers and tickets.

**Entity-Reasoning.** In Entity-Reasoning, we begin by defining subdomains for various entities, such as celebrities, artifacts, and natural landscapes. We manually create several example prompts along with their explicit meanings to guide an LLM in generating more pairs of prompt and its explicit meaning. After collecting 200 such pairs, we carefully review them to ensure overall consistency and confirm that each entity possesses unique visual features. For instance, the prompt "The first mammal successfully cloned from an adult somatic cell in 1996" refers to Dolly the sheep.

**Scientific-Reasoning.** The prompts in the Scientific-Reasoning are constructed in a similar manner. We first identify four key scientific disciplines: physics, chemistry, biology, and astronomy, then create several example pairs of prompt and corresponding explicit meaning. We use these examples to inspire the LLM to generate additional pairs of prompt and explicit meaning. Each prompt is manually validated to ensure it requires reasoning about scientific knowledge and the expected visual outcome is not explicitly stated. For instance, the prompt "A trampoline with an iron ball on it" implies that the heavy iron ball would deeply stretch the surface of the trampoline due to its weight.

Figure 2 demonstrates the prompt collection process (left), shows the subcategories in each dimension (middle) and provides the prompt suite statistics (right). We visualize the word distribution in Figure 4 in Appendix. For more information about the prompt suite, please refer to Appendix A.

## 4 EVALUATION

### 4.1 EVALUATION METRIC

In recent years, MLLMs have demonstrated remarkable capabilities in understanding complex visual content, becoming the primary tool for evaluating images and videos. However, the prompts in our benchmark are highly complex, often involving multiple objects, intricate relationships, and challenging scenarios. As a result, using generic evaluation instructions that are identical for all prompts proved ineffective. This is because each image, generated from a unique prompt, has specific features that require targeted checks. Generic instructions cannot cover every detail, and MLLMs struggle to address all aspects when given long, broad guidelines. To address this, we develop a two-stage evaluation framework with customized evaluation instructions for each prompt. These instructions take into account the prompt category, the reasoning needed, and both the explicit content and implicit meaning the image should exhibit. Figure 3 illustrates the evaluation process.

**Prompt-specific question-criterion pairs generation.** In the first stage, we use the an LLM to generate question-criterion pairs based on the given prompt and dimension-specific information (e.g., idiom meaning for Idiom Interpretation or explicit meaning for Entity and Scientific-Reasoning). For each dimension, two sets of questions are provided to separately examine the reasoning required and the image quality. For Entity and Scientific-Reasoning, where prompts may involve explicit details that do not need reasoning, an additional set of questions is provided to examine these details.

**Image analysis and evaluation.** In the second stage, we employ an MLLM to evaluate the generated images with a Chain-of-Thought (Wei et al., 2022) (CoT) mechanism: the model first describes the image, then answers the specific questions posed in Stage 1. For each question, the MLLM provides an analysis prior to assigning a score, ensuring thorough and reasoned evaluation. Scores within each set are averaged to produce a final metric, which we call 'T2I-ReasonScore'.

$$S_{reason} = \frac{\sum_{i=1}^{n_r} score_i}{n_r}, \tag{1}$$

$$S_{detail} = \frac{\sum_{i=1}^{n_d} score_i}{n_d}, \tag{2}$$

$$S_{quality} = \frac{\sum_{i=1}^{n_q} score_i}{n_q}, \tag{3}$$

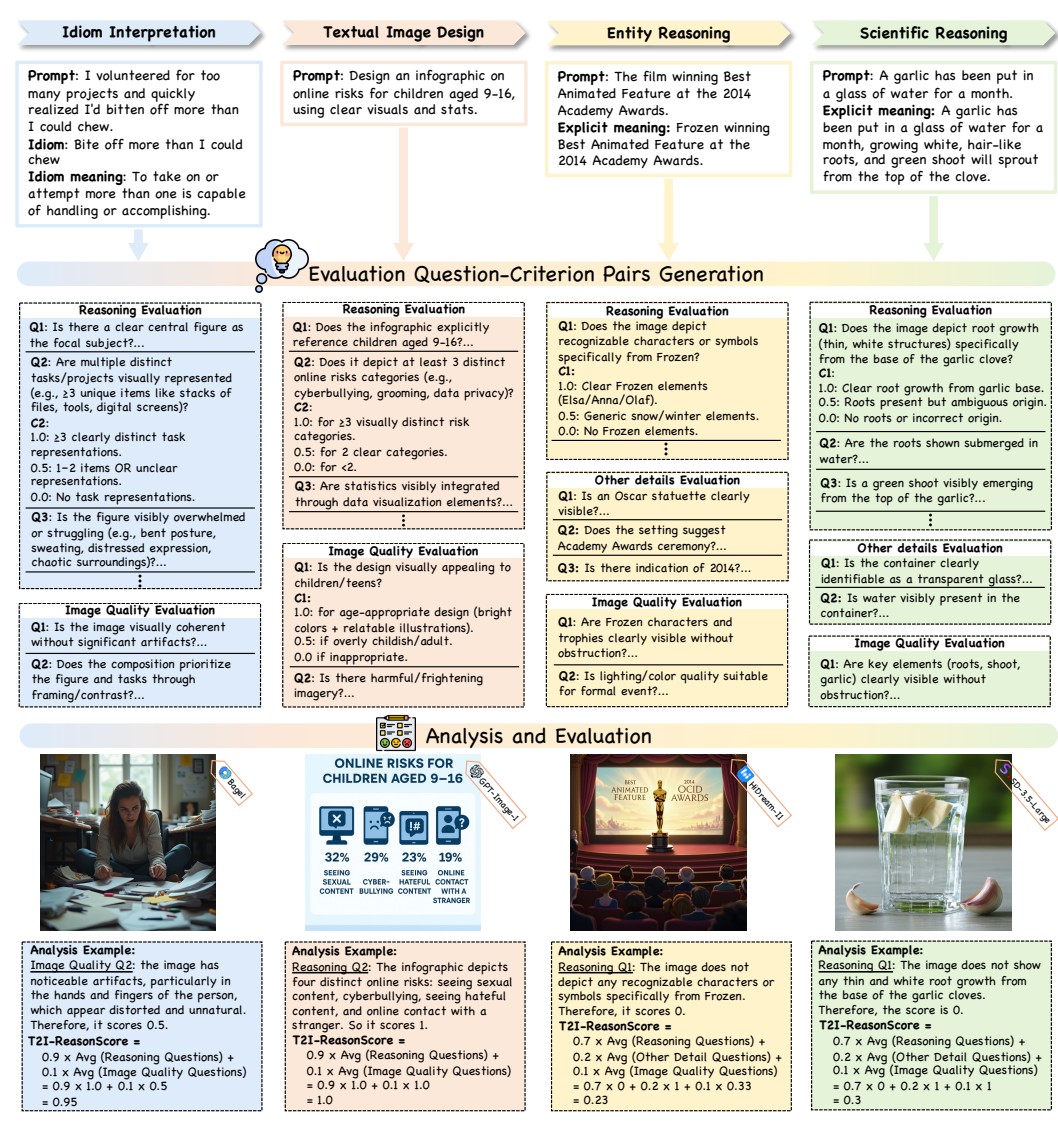

Figure 3: **Evaluation Framework of T2I-ReasonBench.** We adopt a two-stage evaluation framework: prompt-specific evaluation question-criterion pairs generation by an LLM, then image analysis and scoring by an MLLM. This figure shows one evaluation example for each dimension.

where $n_r$, $n_d$, and $n_q$ represent the number of questions in reasoning evaluation, other details evaluation and image quality evaluation.

$$\text{T2I-ReasonScore} = w_1 S_{reason} + w_2 S_{detail} + w_3 S_{quality}, \qquad (4)$$

Here, we set the weights $[w_1, w_2, w_3]$ to $[0.9, 0.0, 0.1]$ for Idiom Interpretation and Textual Image Design, and $[0.7, 0.2, 0.1]$ for Entity-Reasoning and Scientific Reasoning to prioritize reasoning while maintaining a balanced final score.

In this way, our evaluation metrics reflect the reasoning challenges and provide a comprehensive assessment. For more details of our evaluation framework, please refer to Appendix C.

## 4.2 HUMAN EVALUATION CORRELATION ANALYSIS

To validate the effectiveness of our evaluation metric 'T2I-ReasonScore', we perform human evaluations and measure the correlation between our metric and human scores for each dimension. We

Table 1: **The correlation between automatic evaluation metric and human evaluation.** Our proposed metric 'T2I-ReasonScore' show enhanced performance in Kendall's $\tau$ and Spearman's $\rho$.

| Model | Idiom | | Textual | | Entity | | Scientific | | Average | |
|---|---|---|---|---|---|---|---|---|---|---|
| | $\tau(\uparrow)$ | $\rho(\uparrow)$ | $\tau(\uparrow)$ | $\rho(\uparrow)$ | $\tau(\uparrow)$ | $\rho(\uparrow)$ | $\tau(\uparrow)$ | $\rho(\uparrow)$ | $\tau(\uparrow)$ | $\rho(\uparrow)$ |
| CLIPscore Hessel et al. (2021) | 0.3186 | 0.4348 | 0.5372 | 0.7187 | 0.2732 | 0.3837 | 0.1905 | 0.2657 | 0.3299 | 0.4507 |
| VQAscore Lin et al. (2024) | 0.4091 | 0.5672 | 0.4890 | 0.6590 | 0.4483 | 0.6133 | 0.3698 | 0.4939 | 0.4291 | 0.5834 |
| T2I-ReasonScore (ours) | **0.5540** | **0.7115** | **0.6458** | **0.7896** | **0.6514** | **0.7968** | **0.5673** | **0.7231** | **0.6046** | **0.7553** |

randomly select 20 prompts from each dimension and use eight different T2I models to generate 160 images per dimension. This results in 640 images in total for evaluation. The evaluation is conducted with a group of college postgraduate participants, and the criterion is specific for each dimension. Three annotators independently score each image, and we average their scores for each prompt-image pair. We then calculate the correlation between the averaged human scores and the automatic metric score using Kendall's $\tau$ and Spearman's $\rho$. Our metrics are compared against several widely-used T2I evaluation metrics, including CLIPscore (Hessel et al., 2021) and VQAscore (Lin et al., 2024). The correlation results, shown in Table 1, demonstrate that our proposed metric (T2I-ReasonScore) achieve the highest correlation with human judgments across all dimensions (highlighted in bold). For more details about human evaluation, please refer to Appendix D. For more information about selection of models in our two-stage evaluation framework, please refer to Appendix B.

## 5 EVALUATION RESULTS

### 5.1 EXPERIMENTAL SETUP

**Evaluated models.** We evaluate 16 state-or-the-art T2I models, including 8 diffusion T2I models, 5 unified models, and 3 proprietary models. The diffusion T2I models are HiDream-I1-full (hidream, 2024), FLUX.1-devLabs (2024), FLUX.1-schnell (Labs, 2024), Playground-v2.5Li et al. (2024b), Stable-Diffusion-3-Medium (Esser et al., 2024b), Stable-Diffusion-3.5-Medium (Esser et al., 2024b), and Stable-Diffusion-3.5-Large (Esser et al., 2024b), Qwen-Image (Wu et al., 2025). The unified models are: Bagel (Deng et al., 2025), Emu3 (Wang et al., 2024), Janus-Pro-7B (Chen et al., 2025d), show-o-demo-512 (Xie et al., 2024), and GoT (Fang et al., 2025). The proprietary models are Gemini-2.0 (Team et al., 2023), GPT-Image-1 (OpenAI, 2023), and Nano-Banana (Gemini-2.5-Flash-Image) (Google, 2025).

### 5.2 QUANTITATIVE EVALUATION

Table 2 presents the quantitative evaluation results of T2I-ReasonBench. The results reveal significant limitations of current approaches in handling complex prompt reasoning with diverse types of knowledge. Data in the table shows clear performance distinctions between open-source and proprietary models, as well as between specialized T2I diffusion models and unified multimodal models.

**Leading open-source models still limited by concept mapping.** Qwen-Image (Wu et al., 2025) achieves the highest score for open-source models, using a standard double-stream Multimodal Diffusion Transformer architecture trained on web-scale data with particular emphases on text rendering. HiDream (hidream, 2024) exceeds 60 points by incorporating Llama 3 (Grattafiori et al., 2024) as a text encoder, demonstrating the value of integrating powerful LLMs in T2I generation and utilizing their extensive pretraining knowledge. However, both still rely primarily on concept mapping rather than true reasoning, which hinders them achieving truly high scores.

**Unified multimodal models face transfer challenges.** Although unified multimodal models excel in understanding and reasoning text, most of them do not easily transfer this strength to T2I generation, thus underperforming specialized diffusion models. However, Bagel (Deng et al., 2025) achieves comparable results when in its "Thinking" mode. Its bottleneck-free architecture unifies LLM and diffusion model within a single transformer, enhancing interaction between understanding and generation modules.

Table 2: **Evaluation results of T2I-ReasonBench.** Scores are normalized between 0-100. A higher score indicates better performance. Blue highlights the top score in diffusion models. Yellow highlights the top score in unified multimodal models. **Bold** signifies the highest score of all models.

| Model | Idiom | Textual | Entity | Scientific | Overall |
|---|---|---|---|---|---|
| **Diffusion Models** | | | | | |
| SD-3-Medium Esser et al. (2024b) | 41.7 | 73.9 | 41.3 | 56.6 | 53.4 |
| SD-3.5-Medium Esser et al. (2024b) | 39.6 | 71.8 | 43.2 | 55.7 | 52.6 |
| SD-3.5-Large Esser et al. (2024b) | 44.2 | 74.2 | 43.9 | 59.3 | 55.4 |
| FLUX.1-dev Labs (2024) | 49.3 | 72.1 | 45.7 | 51.8 | 54.7 |
| FLUX.1-schnell Labs (2024) | 47.4 | 77.5 | 45.2 | 58.7 | 57.2 |
| Playground-v2.5 Li et al. (2024b) | 50.8 | 53.9 | 44.7 | 54.4 | 51.0 |
| HiDream-I1-full hidream (2024) | 59.1 | 82.3 | 51.5 | 59.4 | 63.1 |
| Qwen-Image Wu et al. (2025) | 62.7 | 83.1 | 59.2 | 68.2 | 68.3 |
| **Unified Multimodal Models** | | | | | |
| Emu3 Wang et al. (2024) | 39.2 | 47.7 | 35.0 | 44.6 | 41.6 |
| show-o-demo-512 Xie et al. (2024) | 38.9 | 48.4 | 34.4 | 48.3 | 42.5 |
| Janus-Pro-7B Chen et al. (2025d) | 32.7 | 49.4 | 36.4 | 52.4 | 42.7 |
| GoT Fang et al. (2025) | 33.7 | 46.5 | 32.9 | 41.3 | 38.6 |
| Bagel w/ Thinking Deng et al. (2025) | 50.6 | 59.0 | 51.9 | 64.9 | 56.6 |
| **Proprietary Models** | | | | | |
| Gemini-2.0 Team et al. (2023) | 65.5 | 83.1 | 68.7 | 76.3 | 73.4 |
| GPT-Image-1 OpenAI (2023) | 84.1 | 94.2 | 76.6 | 82.9 | 84.4 |
| Nano-Banana Google (2025) | **89.8** | **95.4** | **78.8** | **86.0** | **87.5** |

**Proprietary models demonstrate superior reasoning.** Among proprietary models, Gemini-2.0 (Team et al., 2023) demonstrates clear reasoning capabilities by first analyzing prompts before planning visual content. For example, when prompted with "The city hosting the Summer Olympics in 2021", it provides explicit reasoning about Tokyo's landmarks and Olympic imagery. While the technical details of GPT-Image-1 (OpenAI, 2023) remain unpublished, it likely employs a hybrid auto-regressive architecture with a diffusion head, contributing to its strong performance. Leveraging Gemini's extensive world knowledge, Google's Gemini-2.5 Flash Image (Google, 2025) (Nano-Banana) shows even better results.

Figure 5 in Appendix shows more qualitative examples from the evaluated T2I models.

## 5.3 EVALUATION ON TWO-STAGE PIPELINE SETTING

We conduct an additional experiment using a pipeline that decouples reasoning from image generation. In this setup, GPT-4o (Hurst et al., 2024) first reasons about the original prompt and converts it into a visually explicit description, which is then fed to a T2I model. Table 3 presents the quantitative results of using the LLM-rewritten prompts.

**Reasoning is the bottleneck for model performance.** The pipeline setting substantially improves reasoning accuracy for almost all models. This indicates that previous performance gaps between models mainly stem from differences in reasoning abilities. For example, among diffusion-based models, Flux (Labs, 2024) and Stable Diffusion (Esser et al., 2024b) show the most significant improvements, with their performance becoming comparable to HiDream (hidream, 2024). All these models fall within a score interval of 3.0, indicating similar generation capabilities when given clear, direct prompts.

**Internal vs. External Reasoning Abilities.** In this pipeline setting, the "Thinking" mode of Bagel (Deng et al., 2025) is disabled. It shows a substantial overall increase in T2I-ReasonScore, indicating that the external expert LLM has stronger reasoning abilities than its internal understanding module.

Table 3: **Evaluation results of T2I-ReasonBench.** Scores are normalized between 0-100. A higher score indicates better performance. Blue highlights the top score in diffusion models. Yellow highlights the top score in unified multimodal models. **Bold** signifies the highest score of all models.

| Model | Idiom | Textual | Entity | Scientific | Overall |
|---|---|---|---|---|---|
| **Diffusion Models** | | | | | |
| SD-3-Medium | $76.7_{\uparrow 34.9}$ | $82.3_{\uparrow 8.3}$ | $69.0_{\uparrow 27.6}$ | $68.5_{\uparrow 11.9}$ | $74.1_{\uparrow 20.7}$ |
| SD-3.5-Medium | $76.4_{\uparrow 36.8}$ | $82.0_{\uparrow 10.2}$ | $67.9_{\uparrow 24.8}$ | $69.5_{\uparrow 13.9}$ | $74.0_{\uparrow 21.4}$ |
| SD-3.5-Large | $76.7_{\uparrow 32.5}$ | $82.8_{\uparrow 8.6}$ | $72.6_{\uparrow 28.6}$ | $71.4_{\uparrow 12.1}$ | $75.9_{\uparrow 20.4}$ |
| FLUX.1-dev | $77.8_{\uparrow 28.5}$ | $82.1_{\uparrow 10.0}$ | $69.5_{\uparrow 23.8}$ | $71.6_{\uparrow 19.9}$ | $75.3_{\uparrow 20.5}$ |
| FLUX.1-schnell | $79.9_{\uparrow 32.5}$ | $82.8_{\uparrow 5.3}$ | $68.6_{\uparrow 23.4}$ | $74.4_{\uparrow 15.7}$ | $76.4_{\uparrow 19.2}$ |
| Playground-v2.5 | $65.3_{\uparrow 14.5}$ | $62.1_{\uparrow 8.2}$ | $67.7_{\uparrow 22.9}$ | $57.3_{\uparrow 2.8}$ | $63.1_{\uparrow 12.1}$ |
| HiDream-I1-full | $77.0_{\uparrow 17.9}$ | $85.7_{\uparrow 3.4}$ | $73.0_{\uparrow 21.5}$ | $71.6_{\uparrow 12.1}$ | $76.8_{\uparrow 13.7}$ |
| Qwen-Image | $84.7_{\uparrow 22.0}$ | $87.6_{\uparrow 4.5}$ | $78.1_{\uparrow 18.8}$ | $83.7_{\uparrow 15.5}$ | $83.5_{\uparrow 15.2}$ |
| **Unified Multimodal Models** | | | | | |
| Emu3 | $67.7_{\uparrow 28.5}$ | $62.4_{\uparrow 14.7}$ | $59.9_{\uparrow 24.9}$ | $57.3_{\uparrow 12.8}$ | $61.8_{\uparrow 20.2}$ |
| show-o-demo-512 | $74.8_{\uparrow 35.8}$ | $60.6_{\uparrow 12.2}$ | $64.1_{\uparrow 29.8}$ | $65.2_{\uparrow 16.9}$ | $66.2_{\uparrow 23.7}$ |
| Janus-Pro-7B | $72.6_{\uparrow 39.9}$ | $69.9_{\uparrow 20.5}$ | $67.1_{\uparrow 30.7}$ | $68.6_{\uparrow 16.2}$ | $69.5_{\uparrow 26.8}$ |
| GoT | $62.3_{\uparrow 28.7}$ | $53.9_{\uparrow 7.5}$ | $50.3_{\uparrow 17.4}$ | $50.2_{\uparrow 8.9}$ | $54.2_{\uparrow 15.6}$ |
| Bagel w/o Thinking | $77.9_{\uparrow 27.4}$ | $75.5_{\uparrow 16.5}$ | $67.7_{\uparrow 15.8}$ | $74.0_{\uparrow 9.1}$ | $73.8_{\uparrow 17.2}$ |
| **Proprietary Models** | | | | | |
| Gemini-2.0 | $80.3_{\uparrow 14.8}$ | $86.3_{\uparrow 3.1}$ | $77.4_{\uparrow 8.8}$ | $82.5_{\uparrow 6.3}$ | $81.6_{\uparrow 8.2}$ |
| GPT-Image-1 | $87.1_{\uparrow 3.1}$ | $90.3_{\downarrow 3.8}$ | $\mathbf{81.4}_{\uparrow 4.9}$ | $\mathbf{87.3}_{\uparrow 4.4}$ | $86.6_{\uparrow 2.2}$ |
| Nano-Banana | $\mathbf{87.7}_{\downarrow 2.2}$ | $\mathbf{93.6}_{\downarrow 1.8}$ | $81.0_{\uparrow 2.2}$ | $86.5_{\uparrow 0.6}$ | $\mathbf{87.2}_{\downarrow 0.3}$ |

When evaluated with the LLM-rewritten prompts, GPT-Image-1 (OpenAI, 2023) shows only a slight improvement. While Nano-Banana (Google, 2025) maintains the highest overall score, its performance decreases slightly compared to its score with the original, implicit prompts. We interpret these results as follows: for these highly capable models, the original implicit prompts require internal reasoning to determine appropriate visual content. The explicit, pre-reasoned prompts circumvent this need, then the models primarily focus on following the details in prompts. This suggests that the internal reasoning module of GPT-Image-1 is comparable to our external LLM, while that of Nano-Banana can be superior.

**Trade-off Between Reasoning and Instruction-Following.** Interestingly, both the scores of GPT-Image-1 (OpenAI, 2023) and Nano-Banana (Google, 2025) in Textual Image Design decrease slightly. By comparing images, we see that with original concise prompts, the model generates more creative content, whereas the detailed, rewritten prompts constrain it to depict only what is explicitly described. As mentioned previously, this occurs because the models shift their focus to precise instruction-following when given explicit pre-reasoned prompts. This limitation is likely further exacerbated by a potential mismatch between the verbose format of these rewritten design prompts and the models' training data, which additionally constrain their inherent generative capabilities.

**Implication for future model design.** The fact that an integrated model like Nano-Banana (Google, 2025) outperforms the two-stage pipeline suggests that models with inherent, built-in reasoning capacity are superior to a decoupled approach and represent a promising future trend for development.

## 6 CONCLUSION

In this study, we introduce T2I-ReasonBench, a novel benchmark designed to evaluate the reasoning capabilities of T2I generative models. Our evaluation of 16 state-of-the-art T2I models reveals that open-source models have significant limitations in reasoning ability. While proprietary models demonstrate stronger reasoning and knowledge integration, there is still considerable room for improvement.

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

# APPENDIX

## A    MORE DETAILS ON PROMPT COLLECTION PROCESS

**Idiom Interpretation.** In idiom collection, we leverage a book titled "The Exhaustive List of American Idioms" idi (2023), which systematically documents over 11k idioms. These idioms were collected from diverse sources, including TV shows, movies, and everyday conversations. Each idiom in the book is accompanied by its actual meaning in context. In addition to this resource, we also refer to idioms available on the Internet. From this extensive pool, we manually select 200 idioms that are commonly used in daily life and challenging for T2I models due to their figurative meanings. We input the selected idioms and their actual meanings into an LLM and prompt it to generate new sentences. These sentences are designed to describe visible scenes involving the idioms, providing contextual clues for reasoning while avoiding directly revealing the idiom's meaning.

**Textual Image Design.** For textual image design, we collect 6 types of text-rich images from 4 distinct sources.

(1) LLAVAR-2 Dataset Zhou et al. (2024): This dataset contains 42k text-rich images sourced from LAION Schuhmann et al. (2022), representing various categories such as quotes, memes, book covers, posters, and product packaging. However, images in this dataset are of various quality and formats, so we filter out 80 aesthetically pleasing images that have a resolution greater than 384x384 and exhibit clear design intentions.

(2) InfographicVQA Dataset Mathew et al. (2022): This dataset comprises 5k high-quality infographics. We select 40 with normal height-width ratio that exemplify well-crafted layouts to convey structured information.

(3) POSTA Dataset Chen et al. (2025a): This dataset includes over 300 posters with professional background, layout, and text formats designed by experts. We select 40 posters that demonstrate a balance between text and visual design elements.

(4) CoSyn-400k Dataset Yang et al. (2025): This dataset consists of 400k synthetic text-rich images, generated by LLM-drive codes. These images cover diverse formats, such as charts, diagrams, tables, documents (e.g., menus or business cards), math examples, and musical scores. From this dataset, we select 40 samples that exemplify structured text-visual integration, including 10 tables, 10 diagrams, and 20 documents.

The design intentions of the collected images are summarized using Qwen2.5-VL Bai et al. (2025b), yielding 200 prompts in this dimension.

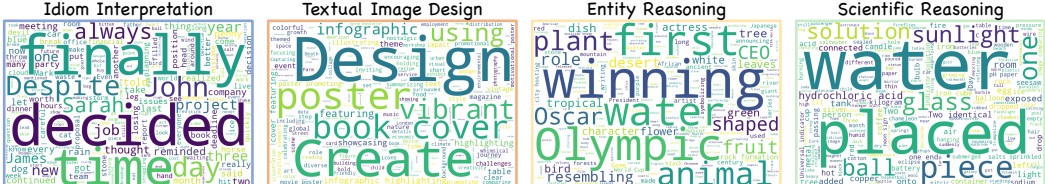

Figure 4: Word cloud to visualize the word distribution of each dimension in our prompt suite.

## B    SELECTION OF EVALUATION MODELS

To address concerns regarding the reliability of our two-stage evaluation framework, specifically, potential biases in the LLM-generated questions and criteria and the limitations of MLLMs, we systematically investigate the robustness of our evaluation metric across a diverse set of models.

### B.1    HUMAN CORRELATION ANALYSIS

This analysis aims to quantify the sensitivity of the scores to specific model choices and to identify optimal, reliable model combination for each reasoning dimension. To this end, we conduct exper-

iments using 12 distinct LLM-MLLM combinations, including 3 LLMs (DeepSeek-R1 Guo et al. (2025), GPT-5.1 OpenAI (2025), and Gemini-2.5-pro Deepmind (2025)) for generating question-criterion pairs and 5 MLLMs (Qwen2.5-VL-72B Bai et al. (2025b), GPT-5.1 OpenAI (2025), LLaVA-OneVision-1.5 An et al. (2025), Gemini-2.5-pro Deepmind (2025), and Qwen3-VL-235B-A22B Bai et al. (2025a)) for the image evaluation stage. The human correlation results for all 12 pipelines are presented in Table 4.

Our quantitative analysis reveals several key insights:

1. Idiom Interpretation: GPT-5.1 consistently outperforms other LLMs in generating effective question-criterion pairs. For the evaluation stage, Qwen3-VL, GPT-5.1, and Gemini-2.5-pro demonstrate comparably strong performance.

2. Textual Image Design: Both GPT-5.1 and Gemini-2.5-pro excel as question generators for this dimension. The Qwen series and GPT-5.1 prove to be particularly adept at evaluating these integrated designs rich in text.

3. Entity Reasoning: Gemini-2.5-pro is the most effective question generator. For evaluation, both Gemini-2.5-pro and GPT-5.1 perform well at recognizing and assessing specific entities.

4. Scientific Reasoning: DeepSeek-R1 and GPT-5.1 generate better questions compared to Gemini-2.5-pro. During image evaluation, the Qwen series and Gemini-2.5-pro perform well, but the combination of GPT-5.1 (LLM) and Gemini-2.5-pro (MLLM) yields the highest correlation.

A clear finding is that different models possess distinct expertise. Overall, the commercial models GPT-5.1 and Gemini-2.5-pro outperform open-source alternatives in both generating evaluation question-criteria pairs and assessing images.

Although it is plausible to select the model combination with highest correlation for each dimension, this is impractical due to the need for frequent model switching. Furthermore, small differences in correlation scores may simply arise from statistical variation rather than meaningful performance gaps. To balance accuracy with practicality, we carefully analyze the model performance consistency across our tests, then select the following model combinations, as highlighted in the table:

1. Idiom Interpretation & Scientific Reasoning: GPT-5.1 (LLM) + Gemini-2.5-pro (MLLM)
2. Textual Image Design & Entity Reasoning: Gemini-2.5-pro (LLM) + GPT-5.1 (MLLM)

We fully acknowledge that the AI field evolves rapidly. Therefore, we will periodically update the correlation rankings with new LLM and MLLM combinations. This will ensure our benchmark remains a credible, transparent, and up-to-date resource, allowing researchers to select evaluation models based on their computational resources and budgets.

## C  EVALUATION FRAMEWORK

We adopt GPT-5.1 OpenAI (2025) and Gemini-2.5-pro Deepmind (2025) as our evaluation tools for different dimensions due to their state-of-the-art performance in visual-textual grounding and fine-grained object recognition. The evaluation of T2I models on our benchmark focuses on two key aspects: reasoning accuracy and image quality. To assess this, we firstly generate specific pairs of question and criterion for each prompt with LLMs.

Table 5, 6, 7, and 8 present the templates used to generate the prompt-specific question-criterion pairs for Idiom Interpretation, Textual Image Design, Entity-Reasoning and Scientific-Reasoning, respectively. Each template is tailored to focus on the unique aspects of its corresponding dimension.

Table 9 presents the template used to evaluate the generated images for all four dimensions. Only the evaluation question-criterion pairs need to be replaced for each prompt.

## D  HUMAN EVALUATION

The human evaluation is conducted on eight models: Stable-Diffusion-3-Medium, FLUX.1-schnell, HiDream-I1-full, Qwen-Image, Bagel, Janus-Pro-7B, GPT-Image-1, and Nano-Banana (4 diffusion

Table 4: **The correlation between automatic evaluation metrics and human evaluation.** We use different LLM-MLLM combinations to calculate 'T2I-ReasonScore'. For each dimension, **Bold** signifies the highest correlation, underline signifies the second highest correlation. We highlight the combination adopted for each dimension in yellow .

| LLM | MLLM | Idiom | | Textual | | Entity | | Scientific | |
|-----|------|----------|----------|----------|----------|----------|----------|----------|----------|
| | | $\tau(\uparrow)$ | $\rho(\uparrow)$ | $\tau(\uparrow)$ | $\rho(\uparrow)$ | $\tau(\uparrow)$ | $\rho(\uparrow)$ | $\tau(\uparrow)$ | $\rho(\uparrow)$ |
| Deepseek | Qwen2.5-VL | 0.5095 | 0.6792 | 0.6051 | 0.7686 | 0.4767 | 0.6242 | 0.4984 | 0.6438 |
| Deepseek | GPT-5.1 | 0.4943 | 0.6659 | 0.6052 | 0.7493 | 0.5432 | 0.6831 | 0.4704 | 0.6168 |
| Deepseek | LLaVA-1V | 0.4664 | 0.6202 | 0.5561 | 0.6936 | 0.5116 | 0.6377 | 0.4342 | 0.5802 |
| GPT-5.1 | Qwen2.5-VL | 0.5391 | 0.7004 | 0.6372 | 0.8002 | 0.5937 | 0.7431 | 0.5062 | 0.6592 |
| GPT-5.1 | GPT-5.1 | 0.5576 | 0.7168 | 0.6446 | 0.8076 | 0.6077 | 0.7642 | 0.4589 | 0.6098 |
| GPT-5.1 | LLaVA-1V | 0.5165 | 0.6594 | 0.5343 | 0.6619 | 0.5762 | 0.7074 | 0.3952 | 0.5130 |
| GPT-5.1 | Gemini-2.5-pro | 0.5540 | 0.7115 | 0.6175 | 0.7848 | 0.6102 | 0.7605 | 0.5673 | 0.7231 |
| GPT-5.1 | Qwen3-VL | 0.5596 | 0.7257 | 0.6324 | 0.7913 | 0.5611 | 0.7114 | 0.5285 | 0.6778 |
| Gemini-2.5-pro | Qwen2.5-VL | 0.4822 | 0.6320 | 0.6829 | 0.8224 | 0.6344 | 0.7882 | 0.4245 | 0.5641 |
| Gemini-2.5-pro | GPT-5.1 | 0.5218 | 0.6672 | 0.6458 | 0.7896 | 0.6514 | 0.7968 | 0.4343 | 0.5781 |
| Gemini-2.5-pro | Gemini-2.5-pro | 0.4975 | 0.6566 | 0.6096 | 0.7641 | 0.6390 | 0.7830 | 0.4471 | 0.5918 |
| Gemini-2.5-pro | Qwen3-VL | 0.4693 | 0.6132 | 0.6527 | 0.7947 | 0.6071 | 0.7399 | 0.5109 | 0.6545 |

models, 2 unified multimodal models, 2 proprietary model). We randomly select 20 prompts from each dimension and use these eight models to generate 160 images per dimension. This results in 640 images in total for human evaluation. The evaluation is conducted with a group of college postgraduate participants. Three participants independently score each image, and we average their scores for each prompt-image pair. The evaluation criteria provided to the human annotators are shown in Table 10. These criteria are tailored to each dimension in order to address different aspects. The correlations are calculated between the averaged human scores and the automatic metric score using Kendall's $\tau$ and Spearman's $\rho$. The results are shown in Table 1.

# E QUALITATIVE EXAMPLES

<USER>: I have a text-to-image generation model that can generate images based on given prompts. However, the model is not perfect and may fail to accurately capture the meaning of the prompt or depict it correctly. Your task is to evaluate the generated image based on a specific prompt that contains an idiom. Given the prompt: {'id': {prompt_id}, 'prompt': {prompt}, 'idiom': {idiom}, 'idiom_meaning': {idiom_meaning}}, you need to:
1. identify what should be depicted in the image or the meaning the image should convey.
2. analyze the prompt and create a list of questions based on the key elements that the image should be checked against.
3. consider factors that could impact the aesthetics or visual quality of the image and list relevant questions.
Please also design a scoring criterion for each question, where a score of 1 means "yes (to the question)", 0 means "no", and 0.5 means "partially yes".
Provide your answer in json format: {{'id': [prompt id], 'prompt': [the prompt], 'image_content': [what the image should convey], 'reason_evaluation': (here should be a dictionary with 3-5 pairs of question and criterion: 'q1': [question 1], 'c1': [criterion 1], 'q2': [question 2], 'c2': [criterion 2]...), 'quality_evaluation': (same format as 'reason_evaluation' with 1-3 pairs of question and criterion)}}.
<ASSISTANT>:
{ "id": 1, "prompt": "He told a funny joke to break the ice at the start of the meeting.",
"image_content": "The image should depict a person telling a joke in a meeting setting, with the audience showing visible signs of relaxation and social easing (e.g., smiling, laughing, or beginning to converse) as a result. It must convey the metaphorical meaning of 'breaking the ice'—reducing tension and initiating interaction—without literal depictions of ice or breaking.",
"reason_evaluation": {
"q1": "Does the image show a meeting environment (e.g., conference table, office setting, or group in professional attire)?",
"c1": "1: Clear meeting setting; 0.5: Ambiguous setting (e.g., generic group); 0: No meeting context",
"q2": "Is a person visibly acting as the joke-teller (e.g., speaking, gesturing, or commanding group attention)?",
"c2": "1: Clear speaker with active role; 0.5: Speaker present but passive; 0: No identifiable speaker",
"q3": "Do audience members display relaxed/post-icebreaking reactions (e.g., genuine smiles, laughter, or eased body language)?",
"c3": "1: Clear reactions showing tension relief; 0.5: Mild reactions (e.g., slight smiles); 0: No reactions or negative expressions",
"q4": "Is the idiom depicted metaphorically (avoiding literal ice/breaking)?",
"c4": "1: Entirely metaphorical; 0.5: Partial literal elements (e.g., ice visuals but not dominant); 0: Literal depiction (e.g., shattered ice)" },
"quality_evaluation": {
"q1": "Is the image visually clear with well-defined subjects (not blurry or distorted)?",
"c1": "1: High clarity; 0.5: Minor blur/distortion; 0: Major quality issues",
"q2": "Is the composition balanced (e.g., speaker and audience arranged cohesively, no awkward cropping)?",
"c2": "1: Harmonious composition; 0.5: Minor imbalances; 0: Chaotic or disjointed layout",
"q3": "Are colors/lighting appropriate and enhance the mood (e.g., warm tones for positivity, no visual clashes)?",
"c3": "1: Lighting/colors support the scene; 0.5: Partially fitting; 0: Inconsistent or jarring"}
}

Table 5: **Template used by DeepSeek-R1 to generate evaluation question-criterion pairs for the dimension of Idiom Interpretation and example output.** The text highlighted in cyan should be replaced with details from the specific prompt.

<USER>: I have a text-to-image generation model that can generate images based on given prompts. However, the model is not perfect and may fail to accurately reflect the prompt or depict the details correctly. Given a prompt which is a design intention for a text-rich image like infographic or poster, your task is to evaluate whether the generated image correctly fulfill the design intention.

Here is the prompt: {{'id': {prompt_id}, 'prompt': {prompt}}}, you need to:

1. identify what should be depicted in the image and its functional purposes.

2. analyze the design intention and create a list of questions based on the key elements that the image should be checked against, including presence of required text elements.

3. consider factors that could impact the aesthetics or visual quality of the image and list relevant questions.

Please also design a scoring criterion for each question, where a score of 1 means "yes (to the question)", 0 means "no", and 0.5 means "partially yes".

Provide your answer in json format: {{'id': [prompt id], 'prompt': [the prompt], 'image_content': [what the image should convey], 'reason_evaluation': (here should be a dictionary with 3-5 pairs of question and criterion: 'q1': [question 1], 'c1': [criterion 1], 'q2': [question 2], 'c2': [criterion 2]...), 'quality_evaluation': (same format as 'reason_evaluation' with 1-3 pairs of question and criterion)}}.

Table 6: **Template used by DeepSeek-R1 to generate evaluation question-criterion pairs for the dimension of Textual Image Design.** The text highlighted in cyan should be replaced with details from the specific prompt.

<USER>: I have a text-to-image generation model that can generate images based on given prompts. However, the prompts given to the model may contain implicit meanings or entities that are not directly stated. Your task is to evaluate whether the generated image accurately represents the intended meaning of the prompt. Given the prompt: {{'id': {prompt_id}, 'prompt': {prompt}, 'explicit_meaning': {explicit_meaning}}}, you need to:

1. identify what should be depicted in the image in order to fully and accurately reflect the explicit meaning of the prompt.

2. identify the entity that the model needs to infer from the prompt, and create a list of questions that check whether the image has correctly identified and depicted this entity.

3. Consider other elements or details in the prompt (apart from the implicit entity), create a list of questions that check if the image accurately reflects these additional key elements.

4. consider factors that could impact the aesthetics or visual quality of the image and list relevant questions.

Please also design a scoring criterion for each question, where a score of 1 means "yes (to the question)", 0 means "no", and 0.5 means "partially yes".

Provide your answer in json format: {{'id': [prompt id], 'prompt': [the prompt], 'explicit_meaning': [the explicit meaning], 'image_content': [what the image should depict], 'entity_evaluation': (here should be a dictionary with 1-3 pairs of question and criterion: 'q1': [question 1], 'c1': [criterion 1], 'q2': [question 2], 'c2': [criterion 2]...), 'other_details_evaluation': (same format as 'entity_evaluation' with 1-3 pairs of question and criterion), 'quality_evaluation': (same format as 'entity_evaluation' with 1-3 pairs of question and criterion)}}.

Table 7: **Template used by DeepSeek-R1 to generate evaluation question-criterion pairs for the dimension of Entity-Reasoning.** The text highlighted in cyan should be replaced with details from the specific prompt.

<USER>: I have a text-to-image generation model that can generate images based on given prompts. However, the prompts given to the model imply scientific laws (e.g., physics, chemistry, biology, or astronomy) that can affect how the scene looks without explicit explanation. Your task is to evaluate whether the generated image accurately reflects the scientific law and correctly portrays the resulting scene. Given the prompt: {{'id': {prompt_id}, 'prompt': {prompt}, 'explicit_meaning': {explicit_meaning}}}, you need to:
1. describe what should be depicted in the image in order to fully and accurately reflect the explicit meaning of the prompt.
2. identify any scientific law(s) that the model needs to infer from the prompt, and create a list of questions that check whether the image correctly demonstrates and complies with these scientific laws.
3. consider other elements or details in the prompt that are not directly affected by the scientific law(s), create a list of questions that check if the image accurately represents these additional key elements.
4. consider factors that could impact the aesthetics or visual quality of the image and list relevant questions.
Please also design a scoring criterion for each question, where a score of 1 means "yes (to the question)", 0 means "no", and 0.5 means "partially yes".
Provide your answer in json format: {{'id': [prompt id], 'prompt': [the prompt], 'explicit_meaning': [the explicit meaning], 'image_content': [what the image should depict], 'scientific_evaluation': (here should be a dictionary with 2-4 pairs of question and criterion: 'q1': [question 1], 'c1': [criterion 1], 'q2': [question 2], 'c2': [criterion 2]...), 'other_details_evaluation': (same format as 'scientific_evaluation' with 1-3 pairs of question and criterion), 'quality_evaluation': (same format as 'scientific_evaluation' with 1-3 pairs of question and criterion)}}.

Table 8: **Template used by DeepSeek-R1 to generate evaluation question-criterion pairs for the dimension of Scientific-Reasoning.** The text highlighted in cyan should be replaced with details from the specific prompt.

<USER>: Describe this image.
<ASSISTANT>: The image depicts a group of four people in a meeting setting, with one person standing and the others seated around a table. The standing individual, dressed in a dark suit and light blue shirt, is playfully pointing at the nose of one of the seated individuals, who is smiling broadly...
<USER>: Based on the image and your previous description, answer the following questions: q1, q2, ...
For each question, assign a score of 1, 0.5 or 0 according to the corresponding scoring criteria: c1, c2, ...
Here are the questions and criteria: {accuracy_evaluation_qc}.
Carefully consider the image and each question before responding, then provide your answer in json format: {{"reason": [your detailed reasoning], "score": [s1,s2, ...]}}
<ASSISTANT>: { "reason": [ "q1: The image clearly shows a meeting environment with a conference table, office-like setting, and individuals in professional attire. This aligns well with the criteria for a clear meeting setting.",
"q2: One person is visibly acting as the joke-teller by making a playful gesture towards another person's face, which suggests an active role in engaging the group. This meets the criteria for a clear speaker with an active role.",
"q3: The audience members are displaying relaxed and post-icebreaking reactions, such as genuine smiles and laughter, indicating that the tension has been relieved. This fulfills the criteria for clear reactions showing tension relief.",
"q4: The idiom 'break the ice' is depicted entirely metaphorically through the playful interaction and the text overlay, without any literal elements like actual ice. This satisfies the criteria for an entirely metaphorical depiction." ], "score": [1, 1, 1, 1] }
<USER>: Based on the image and your previous description, answer the following questions: q1, q2, ...
For each question, assign a score of 1, 0.5 or 0 according to the corresponding scoring criteria: c1, c2, ...
Here are the questions and criteria: {quality_evaluation_qc}
Carefully consider the image and each question before responding, then provide your answer in json format: {{"reason": [your detailed reasoning], "score": [s1,s2, ...]}}
<ASSISTANT>: { "reason": [ "The image is visually clear with well-defined subjects. There is no noticeable blur or distortion, allowing for easy identification of the individuals and their actions. Therefore, it scores high on clarity.",
"The composition is balanced, with the speaker positioned centrally and the audience arranged cohesively around the table. There is no awkward cropping, and all elements contribute to a harmonious arrangement.",
"The colors and lighting are appropriate, enhancing the mood of the scene. Warm tones and even lighting create a positive and inviting atmosphere, supporting the lighthearted interaction depicted." ], "score": [1, 1, 1] }

Table 9: **Template used by Qwen2.5-VL to evaluate generated images in T2I-ReasonBench.** The text highlighted in cyan should be replaced with the specific evaluation question and criterion pairs for the given prompt. An example output for Idiom Interpretation is also provided.

Given the prompt "{prompt}", idiom it contains "{idiom}" and idiom meaning "{idiom meaning}", please rate the alignment between the image and the prompt on a scale of 1 to 5 according to the criteria:

5 - The image independently and unambiguously depicts all elements of the prompt (actions, emotions, context, and consequences) without requiring additional text for further explanation. Annotators can instantly recognize the intended message and its nuances.

4 - The image clearly reflects the core idea of the prompt, capturing major elements (e.g., key actions, settings) but may lack subtle details (e.g., context, specific emotions). Annotators can easily connect it to the prompt with minimal effort.

3 - The image partially represents the prompt, focusing on generic aspects (e.g., basic scenario) but missing critical details (e.g., cause-effect relationships, tone, implied consequences). Annotators can only understand the link after reading the prompt and idiom meaning.

2 - The image vaguely or superficially relates to the prompt, with weak or unclear ties to its specifics (e.g., missing context, conflicting tone, wrong elements). Even with the prompt, the connection feels unclear or underdeveloped.

1 - The image contradicts or ignores the prompt's core message (e.g., misrepresenting outcomes, tone, or relationships). Annotators can find it irrelevant or misleading, even with the prompt.

Table 10: **Human Evaluation Criterion for Idiom Interpretation**

Given a prompt describing a design intention for a rich-text image "{prompt}", please rate how well the image reflects the design prompt on a scale of 1 to 5 according to the criteria:

5 – Exemplary Alignment: The image perfectly reflects the design prompt, addressing all specified elements (e.g., text type, visuals, data, tone), delivers the core message clearly, and has no flaws (no errors, coherent emphasis, and non-superficial intentions fully realized).

4 – Good Alignment with Minor Gaps: The image aligns well with the prompt, fulfills core requirements, and conveys the message effectively but has minor oversights (e.g., missing details, slight color/text inconsistency) that do not undermine the overall intent.

3 – Partial Fulfillment: The image captures the general idea and addresses key aspects of the prompt (e.g., correct type, basic message) but overlooks or misrepresents notable details (e.g., incorrect text/data visualization, inconsistent tone) or contains errors affecting clarity.

2 – Superficial Compliance: The image only superficially resembles the prompt's intent (e.g., correct theme but missing critical elements like key visuals, misaligned focus, or unaddressed design implications) and may include distracting errors or inconsistencies.

1 – Mismatched or Incomplete: The image fails to address the prompt's requirements (e.g., wrong image type, missing core message, major design inaccuracies) with pervasive errors, rendering it ineffective or off-topic.

Table 11: **Human Evaluation Criterion for Textual Image Design**

Given the prompt "{prompt}" and the actual entity it indicates "{rewritten prompts}", please rate the alignment between the image and the prompt on a scale of 1 to 5 according to the criteria:

5 - Perfectly alignment: the image faithfully captures all key elements of the prompt (subject, setting, time period, distinguishing features) with no inaccuracies.

4 - Mostly accurate: the image depicts core elements correctly but has minor errors (e.g., slight anachronisms, missing details, or incomplete context).

3 - Partially correct: the image includes some relevant elements but mixes in inaccuracies (e.g., wrong context, missing critical details, or moderate deviations from the prompt).

2 - Weak representation: the image only loosely connected to the prompt, with significant inaccuracies (e.g., wrong subject identity, era or location).

1 - Completely inaccurate: the image fails to reflect the prompt's core theme, details, or context (e.g., unrelated subject, fantasy elements, or contradictory visuals).

Please carefully examine the image and check if all the details in the prompt are correctly addressed in the image.

Table 12: **Human Evaluation Criterion for Entity Reasoning**

Given a prompt that relates to scientific laws "{prompt}", please rate the image on a scale of 1 to 5 according to the criteria:

5 - Excellent: The image accurately depicts all the elements from the prompt (subject, action, setting, state) and strictly adheres to scientific laws. No errors in details or logic.

4 - Good: The image includes all key elements from the prompt but has minor scientific inaccuracies or small missing details.

3 - Fair: The image includes most elements but has moderate errors: either missing an critical element or clearly violating scientific principles.

2 - Poor: The image omits multiple key elements and has significant scientific inaccuracies.

1 - Fail: The image fails to represent the prompt (e.g., incorrect subjects/actions) and completely ignores scientific laws.

Please carefully examine the image and check if the image correctly address the scientific law inherent in the prompt.

Table 13: **Human Evaluation Criterion for Scientific Reasoning**

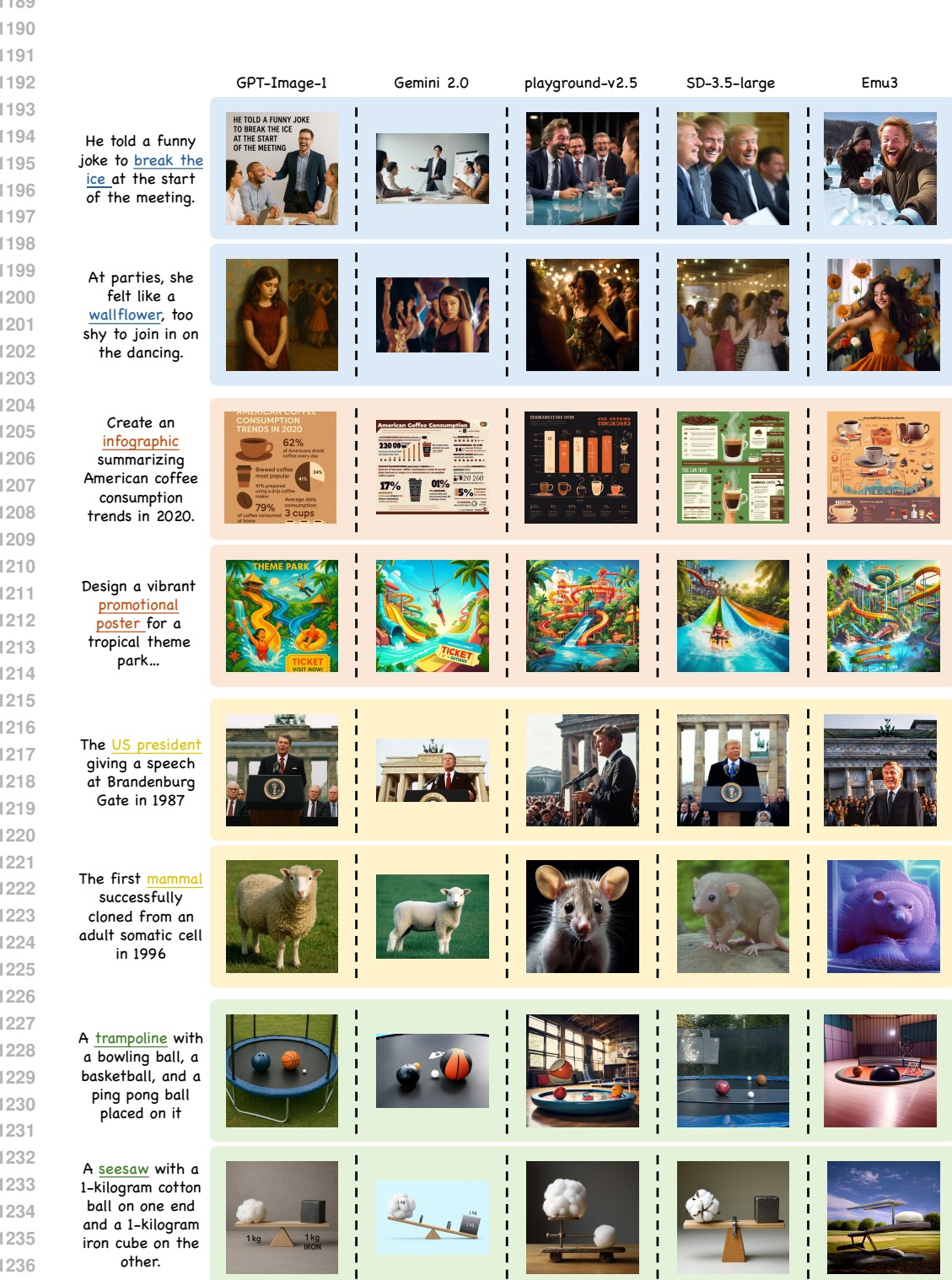

Figure 5: Qualitative examples.

