# OpenReview forum: "T2I-ReasonBench: Benchmarking Reasoning-Informed Text-to-Image Generation"
_ICLR.cc/2026/Conference — ICLR 2026 Conference Withdrawn Submission_

### Official Review · Reviewer_RZgd · 2025-10-27

**Soundness:** 2
**Presentation:** 3
**Contribution:** 3
**Rating:** 4
**Confidence:** 3

**Summary:**

This paper introduces T2I-ReasonBench, a novel benchmark designed to evaluate the reasoning capabilities of text-to-image (T2I) models, moving beyond literal prompt adherence. The benchmark features 800 prompts across four dimensions: idiom interpretation, textual image design, entity reasoning, and scientific reasoning. It employs a two-stage evaluation framework where an LLM generates question-criteria pairs, and an MLLM scores the resulting images. Evaluations of 16 state-of-the-art models show that reasoning, rather than generation, is the primary bottleneck for current systems.

**Strengths:**

1. This work addresses the significant and under-explored problem of evaluating T2I models' reasoning capabilities, moving beyond existing benchmarks that focus on literal prompt-image alignment.
2. The benchmark introduces novel dimensions, "Idiom Interpretation" and "Textual Image Design," which challenge models with complex, abstract tasks that require inferring implicit information rather than just following explicit instructions.
3. Through a comprehensive evaluation of 16 SOTA models and an insightful LLM-rewrite experiment, the study compellingly demonstrates that reasoning is the primary bottleneck for current T2I models and highlights the superiority of integrated reasoning designs.

**Weaknesses:**

1. The benchmark's core "AI-evaluating-AI" evaluation framework is a key weakness, as its reliability depends entirely on the AI models used for evaluation.
a.	If the criteria-generating LLM (DeepSeek-R1) itself possesses biases, knowledge gaps, or reasoning errors, it will produce flawed question-criterion pairs  from the very start.
b.	This pipeline is susceptible to compounding errors, where any biases or misunderstandings from the LLM in the first stage are amplified by the MLLM's own limitations in the second.
The paper lacks an in-depth analysis of this potential evaluation bias.


2. The benchmark's overall size of 800 prompts  is relatively small, which may limit its robustness for a comprehensive evaluation. This limitation is particularly evident in the "Scientific-Reasoning" dimension, which attempts to cover four vast and distinct disciplines—physics, chemistry, biology, and astronomy —with only 200 prompts, which may limit its robustness for a comprehensive evaluation, particularly in the ‘Scientific-Reasoning’ dimension, where coverage seems sparse across diverse disciplines.

3. Several tasks within the benchmark, particularly in the Entity-Reasoning dimension, appear to test factual knowledge retrieval more than inferential reasoning. For example, the prompt "The first mammal successfully cloned from an adult somatic cell in 1996" primarily assesses whether the model knows the answer is "Dolly the sheep". While knowledge is a prerequisite for reasoning, this specific task seems to conflate "knowing a fact" with the more complex process of "reasoning from facts" to produce a visual output.

4. For improved readability, the structure of Section 4 could be slightly reorganized. The paper's primary metric, the $T2I-ReasonScore$, is defined entirely within the section's introductory text, while its subsequent validation is placed in the sole subsection (4.1). This organization slightly de-emphasizes the metric's definition; creating a dedicated subsection for the metric's formulation would provide a more balanced and intuitive flow for the reader.

**Questions:**

1.	In your pipeline experiment (Section 5.3), you present a very interesting finding: using more explicit, LLM-rewritten prompts decreased the 'Textual Image Design' scores for GPT-Image-1 and Nano-Banana. You attribute this to the detailed prompts limiting the models' "creative freedom". This seems to reveal a tension between "reasoning"  and "creativity". How do you view this trade-off?

2.	Could you please clarify the operational definition of 'reasoning' used in this benchmark and explain how you distinguish this from simple fact recall?

3.	Could you provide more detailed statistics about the benchmark, specifically the distribution of prompts across the scientific subcategories (e.g., the number of prompts for physics, chemistry, biology, and astronomy)?

---

> ### Author Response · Authors · 2025-11-26
> **Response to Reviewer RZgd (1/4)**
>
> We sincerely thank the reviewer for the thorough and constructive feedback. Our point-by-point responses to the specific questions are detailed below.
>
> > **Q1: Reliability of AI-evaluating-AI**
>
> **A1**:
> We thank the reviewer for raising this important point regarding the reliability of our evaluation framework.
>
> We would like to clarify that this two-stage evaluation framework, i.e., using VQA models on LLM-generated questions derived from prompts, has been **widely used in T2I evaluation** since the advent of powerful LLMs and MLLMs. It has been successfully employed in several prior works, including TIFA [A], VQ^2 [B], VPEval [C], Davidsonian Scene Graph [D] ConceptMix [E] for text-image alignment, and I-HallA [F] for factual correctness.
>
> - **Step 1: Comparison of human correlations using different LLM and MLLM combinations**:
>
>    To rigorously address the potential for compounding errors and model-specific biases, we conducted **experiments using 2 different LLMs and 3 different MLLMs**, resulting in 6 unique evaluation pipeline configurations. The new LLM is GPT-5.1 (released by OpenAI in November), and the additional MLLMs are GPT-5.1 and LLaVA-OneVision [G]. We first expanded our human evaluation set to include 8 models (+3 more models). The human correlations for all 6 pipelines are presented in **Table 1**.
>
>
>    Our analysis reveals two key findings:
>
>    1. Regarding the Stage 1 LLM: **GPT-5.1 generally outperforms DeepSeek-R1**. When using the same MLLM evaluator, pipelines with question-criterion sets generated by GPT-5.1 achieve higher correlations in almost all dimensions.
>
>    2. Regarding the Stage 2 MLLM: **The open-source model Qwen2.5-VL demonstrates performance comparable to the commercial GPT-5.1**. When provided with questions from GPT-5.1, Qwen2.5-VL's correlation is even higher.
>
> - **Step 2: Comparing question-criterion samples generated by different LLMs**:
>
>    To further investigate the quality of the generated questions & criteria, we conducted a human evaluation on 80 samples from each LLM. We found that both LLMs produced reasonable, non-hallucinated questions and criteria for all prompts, with no instances of reasoning errors or critical knowledge gaps. The questions generated by both models were largely overlapping and addressed the same core aspects of the prompts.
>
>    However, GPT-5.1 demonstrated better quality in two ways:
>
>    1. Its **questions and criteria are more detailed and specific**, with clearer descriptions of expected visual elements.
>
>    2. The **structure is more refined**, with each question being more "atomic" (covering the smallest semantic unit) and exhibiting less overlap between questions within a set.
>
>       For example, for the idiom prompt *"After three weeks of burning the midnight oil, she finally submitted her dissertation"*:
>
>       * GPT-5.1's first two questions are:
>
>          * "q1": "Does the image clearly show a female main character as the central figure?"
>          * "q2": "Is it clear that she is working on or has just completed an academic dissertation or similar substantial written work?"
>
>       * DeepSeek-R1's first two questions are:
>          * "q1": "Does the image clearly show a woman engaged in academic work?" **(combines 'woman' and 'work')**
>          * "q2": "Are academic materials (books/papers/laptop) prominently featured?" **(overlaps with the academic theme of q1)**
>
>
> While GPT-5.1 delivers better performance as stage 1 LLM, it was not available at the time of our initial experiments. A key reason for our original choice of DeepSeek-R1 and Qwen2.5-VL is that they are open-source models, more accessible for the research community.
>
> We fully acknowledge that the AI field evolves rapidly. Therefore, in the final version, we commit to **releasing all human scores and will periodically update the correlation rankings with new LLM and MLLM combinations**. This will ensure our benchmark remains a credible, transparent, and up-to-date resource, allowing researchers to select evaluation models based on their computational resources and budgets.
>
> ---------------------------------------------------------------
> [A] Hu, Yushi, et al. TIFA: Accurate and interpretable text-to-image faithfulness evaluation with question answering. 2023.
>
> [B] Yarom, Michal, et al. What you see is what you read? improving text-image alignment evaluation. 2023
>
> [C] Cho, Jaemin, Abhay Zala, and Mohit Bansal. Visual programming for step-by-step text-to-image generation and evaluation. 2023
>
> [D] Cho, Jaemin, et al. Davidsonian scene graph: Improving reliability in fine-grained evaluation for text-to-image generation. 2023
>
> [E] Wu, Xindi, et al. ConceptMix: A compositional image generation benchmark with controllable difficulty. 2024
>
> [F] Lim, Youngsun, Hojun Choi, and Hyunjung Shim. Evaluating Image Hallucination in Text-to-Image Generation with Question-Answering. 2025
>
> [G] Li, Bo, et al. Llava-onevision: Easy visual task transfer. 2024

---

> ### Author Response · Authors · 2025-11-26
> **Response to Reviewer RZgd (2/4)**
>
> **Table 1. Human Correlations using different LLMs and MLLMs**
> |LLM|MLLM| Idiom | Idiom | Textual | Textual | Entity | Entity | Scientific | Scientific | Average | Average |
> |------------|------------|------------|------------|------------|------------|------------|------------|------------|------------|------------|------------|
> | || τ  | ρ  |  τ  | ρ  | τ  | ρ  | τ  | ρ  | τ  | ρ | τ  | ρ |
> |DeepSeek|Qwen2.5-VL|0.5095|0.6792|0.6051|0.7686|0.4767|0.6242|0.4984|0.6438|0.5224 |**0.6790**
> |DeepSeek|GPT-5.1   |0.4943|0.6659|0.6052 |0.7493|0.5432|0.6831|0.4704|0.6168|**0.5283** |0.6788
> |DeepSeek|LLaVA-1V |0.4664|0.6202|0.5561|0.6936|0.5116|0.6377|0.4342|0.5802|0.4921 |0.6329
> |GPT-5.1 |Qwen2.5-VL|0.5391|0.7004|0.6372|0.8002|0.5937|0.7431|0.5062|0.6592|**0.5691** |**0.7257**
> |GPT-5.1 |GPT-5.1   |0.5576|0.7168|0.6446|0.8076|0.6077|0.7642|0.4589|0.6098|0.5672|0.7246
> |GPT-5.1 |LLaVA-1V |0.5165|0.6594|0.5343|0.6619|0.5762|0.7074|0.3952|0.5130|0.5056 |0.6354

---

> ### Author Response · Authors · 2025-11-26
> **Response to Reviewer RZgd (3/4)**
>
> > **Q2: Clarification on benchmark size and statistics**
>
> **A2**: We apologize for not including the detailed prompt distribution in the original submission.
>
> Here are the specific numbers of prompts per subcategory:
>
> * Scientific-Reasoning: 200
>   * Physics: 100
>   * Chemistry: 50
>   * Biology: 30
>   * Astronomy: 20
>
> * Entity-Reasoning: 200
>   * Celebrity: 49
>   * Artifact: 27
>   * Food: 26
>   * Event: 23
>   * Architecture: 21
>   * Animal: 20
>   * Plant: 18
>   * Nature: 16
>
> Pie chart in Figure 2 are plotted proportionally to the distribution. We will update it to annotate these numbers, providing clearer benchmark statistics.
>
> **Benchmark Size and Composition:**
>
> We agree that the coverage of certain scientific subfields could be expanded. However, unlike benchmarks that evaluate literal text-to-image alignment, T2I-ReasonBench focuses on reasoning-informed generation. This requires each prompt to be **meticulously designed and manually verified to ensure it genuinely tests reasoning capabilities**, making it unsuitable to scale via simple templates. Within each dimension, our primary aim was to cover a broad spectrum of topics to provide a meaningful overview of model abilities, which resulted in the current distribution.
>
> - **Similar size with other benchmarks**:
> When compared to other related benchmarks (as shown in **Table 2**), our prompt count is consistent with the current state of the field. For benchmarks that include reasoning-based prompts, whether focused on a single domain like physics or covering multiple categories, the total number of such prompts typically remains under 1000. This reflects the ongoing, exploratory nature of reasoning-informed T2I evaluation and the current challenge of automatically generating large volumes of high-quality reasoning prompts.
>
> We acknowledge this as a valuable direction for future work and are committed to expanding both the spectrum of reasoning tasks and the number of prompts in the future.
>
>
>
>
> **Table 2. Comparison of Reasoning T2I Benchmarks**
> |Benchmark|Total Prompts| Related Dimensions | Category Prompts| Remarks
> |------------|------------|------------|------------|------------|
> |PhyBench [H]|700|Physics|700|
> |R2I-Bench [I]|3068|Commonsense|695| some prompts relating to entities/physics
> |            |    |Causal|151| some prompts relating to physics
> |WISE [J]        |1000|Cultural Commonsense|400|prompts about entities but easier
> |               |    |Science|300|Physics(100), Chemistry(100), Biology(100)
> |Commonsense-T2I [K]|300 |Physical Laws|98|
> |                  |    |Biological Laws|34|
> |                  |    |Animal Behaviors|36|
> |ours|800 |Entity|200|
> |    |    |Scientific|200|Physics(100), Chemistry(50), Biology(30), Astronomy(20)
>
> ---------------------------------------------------------------
> [H] Meng, Fanqing, et al. PhyBench: A physical commonsense benchmark for evaluating text-to-image models. 2024
>
> [I] Chen, Kaijie, et al. R2I-Bench: Benchmarking Reasoning-Driven Text-to-Image Generation. 2025
>
> [J] Niu, Yuwei, et al. WiSE: A world knowledge-informed semantic evaluation for text-to-image generation. 2025
>
> [K] Fu, Xingyu, et al. Commonsense-T2I challenge: Can text-to-image generation models understand commonsense?. 2024

---

> ### Author Response · Authors · 2025-11-26
> **Response to Reviewer RZgd (4/4)**
>
> > **Q3: Clarifying the Distinction Between Factual Recall and Reasoning in Our Benchmark**
>
> **A3**: We thank the reviewer for this critical question regarding the operational definition of 'reasoning' in our benchmark.
>
> We agree that knowledge is a prerequisite for reasoning. T2I-ReasonBench is designed to create scenarios that require models to go beyond simple fact retrieval by **integrating domain knowledge to infer implicit meaning, resolve contextual ambiguities, and ultimately determine the appropriate visual content through reasoning**.
>
> - **Distinction between knowledge retrieval and reasoning**:
>
>    1. **Knowledge Alone Is Insufficient Without Reasoning**: Even models with extensive knowledge bases can fail our tasks if they lack reasoning capability. For instance:
>       - Diffusion models with LLM as text encoders should possess substantial knowledge, like Qwen-Image (using Qwen2.5-VL as text encoder) and HiDream (using Llama-3.1), yet they generated images of a mouse instead of a sheep when prompted with *"The first mammal successfully cloned from an adult somatic cell in 1996"*.
>       -  This failure demonstrates that merely having knowledge is insufficient; models require **reasoning ability to transfer and integrate that knowledge into image generation**.
>
>
>    2. **Defining Reasoning Ability in T2I Generation:**
>
>       - A key distinction lies in how reasoning is architected. In a two-stage pipeline (LLM rewrite + image generation), the tasks of knowledge retrieval and reasoning are primarily handled by the LLM. In contrast, for Unified Multimodal Models (UMMs), which inherently possess both knowledge and reasoning capacities, the critical challenge lies in **effectively transferring this understanding into the image generation**
>
>       - Therefore, within the context of T2I generation, we define "reasoning" as the **end-to-end capability that encompasses not only knowledge retrieval and logical inference but, crucially, the successful application of that reasoning to produce a semantically faithful image**. A key demonstration of this reasoning ability in UMMs is the generation of a textual or implicit reasoning chain that guides the image generation.
>
> - **Illustrative Examples of Reasoning in T2I**:
>
>    - For *"The country winning the FIFA World Cup in 2014"*, proper reasoning involves generating the German national team celebrating with the trophy in a football context, rather than simply producing an image of Germany's flag or map.
>
>    - The prompt *"The basketball team winning 6 NBA championships in the 1990s"* requires a multi-step reasoning process: first recalling championship data across a decade, then identifying the unique team (Chicago Bulls) that won exactly six times during that period.
>
>    - We observe that even models with knowledge and reasoning ability can fail. Given the prompt *"The Mexican fruit with leathery skin and blood-red pulp containing edible black seeds"* to GPT-Image-1, whose text answer "red dragon fruit" is correct, but still generate a fruit with some correct features but definitely not red dragon fruit, demonstrating a failure in integrating knowledge in generation.
>
> These examples illustrate that our benchmark evaluates the reasoning process through which **models apply knowledge to create contextually appropriate visual representations, rather than testing factual recall in isolation**. We acknowledge that future work could benefit from establishing varying difficulty levels that more explicitly separate knowledge dependency from reasoning complexity.
>
>
> > **Q4: Performance decrease with LLM-rewritten prompts.**
>
> **A4**: We thank the reviewer for this comment regarding the observed performance decrease with LLM-rewritten prompts.
>
> - Our interpretation is that for highly capable models like GPT-Image-1 and Nano-Banana, the original, implicit prompts require them to perform their own internal reasoning to determine the appropriate visual content. In contrast, the **LLM-rewritten prompts are already explicit and detailed, effectively reducing the need for reasoning**.
>
> - When presented with these detailed, pre-reasoned prompts, the models focus on **following the detailed instructions**. This is further compounded by a potential **mismatch between the format of these verbose prompts and the models' training data**, which additionally constrain their natural generative capabilities. We have revised the main text to reflect this more nuanced interpretation of this interpretation.
>
>
> > **Q5: Reorganization of Section 4 for Improved Clarity**
>
> **A5**:
> We sincerely thank the reviewer for this valuable suggestion to improve the paper's organization.
>
> We have restructured Section 4 accordingly by creating a dedicated subsection (4.1) specifically for the formulation and definition of our primary metric, the **T2I-ReasonScore**.
> The human evaluation has been moved to subsection 4.2.

---

### Official Review · Reviewer_jgck · 2025-10-31

**Soundness:** 2
**Presentation:** 3
**Contribution:** 3
**Rating:** 4
**Confidence:** 4

**Summary:**

This paper introduces T2I-ReasonBench, a new benchmark designed to evaluate the reasoning capabilities of text-to-image (T2I) models, moving beyond the literal prompt-following assessments of existing work. The benchmark consists of 800 prompts across four dimensions: Idiom Interpretation, Textual Image Design, Entity-Reasoning, and Scientific-Reasoning. The authors also propose a two-stage automated evaluation framework where an LLM generates prompt-specific criteria and a Multimodal LLM (MLLM) scores the generated images, producing a 'T2I-ReasonScore'. The paper benchmarks 16 state-of-the-art models and concludes that reasoning, rather than image generation fidelity, is the primary bottleneck for current models. A key finding is that performance significantly improves when an external LLM first rewrites implicit prompts into explicit ones.

**Strengths:**

- The paper addresses a timely and critical problem in generative AI: moving beyond surface-level text-image alignment to evaluate the deeper reasoning capabilities of T2I models. This is an important direction for the field.
- The proposed benchmark is reasonably comprehensive, with four distinct dimensions that probe different facets of reasoning, from figurative language (idioms) and creative planning (textual design) to world knowledge (entities) and physical principles (scientific reasoning).
- The experiment using an LLM to rewrite implicit prompts into explicit ones provides a valuable insight. It effectively decouples the reasoning and generation tasks, offering compelling evidence that the reasoning ability of T2I models is a major performance bottleneck.
- The experimental analysis is comprehensive, covering 16 different models. The experiment using a two-stage pipeline (LLM-rewrite + T2I-generate) is particularly insightful, providing strong evidence that reasoning is a major bottleneck for current T2I systems.

**Weaknesses:**

- Potential for evaluator bias: The framework uses Qwen2.5-VL as the automated scorer. Given that Qwen-Image is one of the top-performing open-source models under evaluation, this raises a serious concern about potential 'in-family' bias. The human correlation analysis, conducted on an unspecified subset of only 5 models, is not sufficient to rule out this potential bias across all 16 evaluated models. This concern undermines the reliability of the reported model rankings.
- Arbitrary evaluation metric: The final T2I-ReasonScore is a weighted average of sub-scores, with manually set weights (e.g., [0.7, 0.2, 0.1]). The paper provides no sensitivity analysis to show how the model rankings would change with different weights. This makes the final scores seem arbitrary and potentially not robust.
- Oversimplified interpretation of results: The paper claims that the performance drop of the Nano-Banana model on rewritten prompts implies a 'superior internal reasoning module'. This is an overstatement. A more plausible alternative, which the paper even acknowledges in the context of Textual Image Design, is that the verbose, explicit prompts are simply a poor format for this model's input processor, constraining its creative abilities. This nuance is lost in the main conclusion.
- Reasoning vs. memorization: The benchmark does not sufficiently disentangle genuine reasoning from the retrieval of memorized associations. For idioms and specific entities, it is highly likely that top-performing models are leveraging patterns seen in their massive training datasets rather than performing multi-step reasoning. The paper acknowledges this possibility but does not offer a solution, which challenges the benchmark's core objective.

**Questions:**

- Can you comment on the potential for systemic bias from using Qwen2.5-VL to evaluate Qwen-Image? To strengthen the benchmark's credibility, have you considered cross-validating the model rankings with a powerful, architecturally distinct MLLM (e.g., GPT-4V or a LLaVA variant)?
- Could you provide a sensitivity analysis for the score weights? How much do the model rankings change if the weights for reasoning, detail, and quality are varied? This would help establish the robustness of your findings.

---

> ### Author Response · Authors · 2025-11-23
> **Response to Reviewer jgck (1/5)**
>
> We sincerely thank the reviewer for the thorough and constructive feedback. The insightful comments have been invaluable in helping us strengthen the paper. We have carefully considered all the points raised and have revised the manuscript accordingly. Our point-by-point responses to the specific concerns are detailed below.
>
> > **Q1**: Potential for evaluator bias
>
> **A1**: We thank the reviewer for this important question regarding potential "in-family" bias, as Qwen-Image is evaluated by Qwen2.5-VL. To thoroughly investigate this, we conducted three progressive experiments:
>
> 1. **Expand Human Correlation**:
> We apologize for the lack of detail in the original submission. The human evaluation was initially conducted on five models: Stable-Diffusion-3-Medium, HiDream-I1-full, Bagel, Janus-Pro-7B, and GPT-Image-1 (2 diffusion models, 2 unified multimodal models, 1 proprietary model).
> To ensure broader representativeness, we have now expanded this set to eight models by adding FLUX.1-schnell, Qwen-Image, and Nano-Banana. The results are shown in **Table 1**. The correlation for this expanded set remains highly consistent with our original correlation of 5 models (variations within 0.03). Furthermore, we calculated correlations for 2 different 5-model subsets (subset 2 & 3), 2 different 7-model subsets (subset 4 & 5), one with and one without Qwen-Image. All combinations yielded similar correlations, strongly indicating that **Qwen2.5-VL does not exhibit bias when evaluating Qwen-Image**.
>
> 2. **Cross-Validation with Architecturally Distinct MLLMs**:
> To further verify the effectiveness of Qwen2.5-VL, we re-evaluated the eight models using two powerful and distinct MLLMs: GPT-5.1 (the latest OpenAI model, as GPT-4V no longer available) and LLaVA-OneVision [A].
> The model rankings, change in rankings compared with Qwen2.5-VL, and model scores for each dimension are shown in **Table 2**. The results show that the model rankings are highly consistent across all three evaluators:
>    - For the Idiom and Entity, the rankings are identical.
>    - For Textual Image Design and Scientific, minor rank swaps (+1 or -1) occur only between models whose original scores were very close. These models also have nearly identical scores (within 3 points) under the other MLLMs.
>
>    **This confirms the reliability of Qwen2.5-VL and the rankings reported in our paper are robust.**
>
> 3. **Compare human correlation with GPT-5.1 and LLaVA-OneVision**:
> We also compared the human correlation of all three MLLM evaluators on the eight models, results shown in **Table 3**. This revealed a nuanced insight: Qwen2.5-VL achieved the highest correlations on all dimensions except Entity, likely because it is less adept at recognizing specific facial features of certain celebrities, relying instead on coarser attributes (gender, outfit, environment).
>
>    Critically, **Qwen2.5-VL achieved the highest overall correlation (Spearman’s ρ) with human judgments among the three evaluators**, as an open-source model, it performs comparably to the recently released commercial model GPT-5.1, justifying its use as our MLLM evaluator at this time.
>
> **Conclusion**: These three experiments progressively demonstrate that Qwen2.5-VL introduces no bias in favor of Qwen-Image, and that the benchmark rankings are robust. We acknowledge that AI field evolves rapidly, there will always be more powerful models emerging; therefore, in the final version, we will release all human scores and plan to periodically update correlation rankings with new MLLMs to ensure the benchmark remains credible and transparent.
>
> [A] An, Xiang, et al. "Llava-onevision-1.5: Fully open framework for democratized multimodal training."

---

> ### Author Response · Authors · 2025-11-23
> **Response to Reviewer jgck (2/5)**
>
> **Table 1. Human Correlations on different model subsets**
> | Model Combination | Idiom Interp. | Idiom Interp. | Textual Image | Textual Image| Entity Reasoning | Entity Reasoning | Scientific Reasoning| Scientific Reasoning|
> |------------|------------|------------|------------|------------|------------|------------|------------|------------|
> || τ  | ρ  |  τ  | ρ  | τ  | ρ  | τ  | ρ  | τ  | ρ  |
> |8 models| 0.5095 | 0.6792 | 0.6051 | 0.7686 | 0.4767 | 0.6242 | 0.4984 |0.6438
> |subset 1| 0.5255 | 0.6786 | 0.6087 | 0.7793 | 0.4884 | 0.6353 | 0.4847 | 0.6163 |
> |subset 2| 0.5243 | 0.7045 | 0.6136 | 0.7612 | 0.4870 | 0.6281 | 0.4929 | 0.6473 |
> |subset 3| 0.5031 | 0.6707 | 0.6079 | 0.7720 | 0.4620 | 0.6143 | 0.5017 | 0.6411 |
> |subset 4| 0.5102 | 0.6738 | 0.6180 | 0.7842| 0.4821 | 0.6331 | 0.5024 | 0.6531 |
> |subset 5| 0.5145 | 0.6859 | 0.6217 | 0.7859 | 0.4897 | 0.6386 | 0.4932 | 0.6346 |
> * subset 1 (same as paper):     SD-3-medium, Hidream, Bagel, Janus, and GPT-Image-1
> * subset 2 (w/ Qwen-Image):   Qwen-Image, Flux.1-Schnell, Janus, Nano-Banana, GPT-Image-1
> * subset 3 (w/o Qwen-Image):  Flux.1-Schnell, SD-3-medium, Hidream, Bagel, Nano-Banana
> * subset 4 (w Qwen-Image):     Qwen-Image, SD-3-medium, Flux.1-Schnell, Hidream, Janus, Bagel, Nano-Banana
> * subset 5 (w/o Qwen-Image):  SD-3-medium, Flux.1-Schnell, Hidream, Janus, Bagel, Nano-Banana, GPT-Image-1
>
>
> **Table 2. Model ranking change and scores using different MLLM evaluator**
> |Rank| Idiom | Idiom| Textual | Textual| Entity| Entity| Scientific| Scientific|
> |------------|------------|------------|------------|------------|------------|------------|------------|------------|
> || GPT-5.1 | LLaVA-1V  |  GPT-5.1 | LLaVA-1V | GPT-5.1 | LLaVA-1V | GPT-5.1 | LLaVA-1V | GPT-5.1 | LLaVA-1V |
> |1| Nano-Banana(--): 82.0  | Nano-Banana(--): 83.0  | **GPT-Image-1(+1)**: 98.2  | Nano-Banana(--): 94.8  | Nano-Banana(--): 80.9  | Nano-Banana(--): 83.3  | Nano-Banana(--): 79.9  | Nano-Banana(--):  82.7  |
> |2| GPT-Image-1(--): 74.4  | GPT-Image-1(--): 81.1  | **Nano-Banana(-1)**: 97.0  | GPT-Image-1(--): 91.6  | GPT-Image-1(--): 77.6  | GPT-Image-1(--): 81.0  | GPT-Image-1(--): 76.8  | GPT-Image-1(--):  81.7  |
> |3| Qwen-Image(--):  54.8  | Qwen-Image(--):  62.1  | Hidream(--):     94.1  | **Qwen-Image(+1)**:  80.6  | Qwen-Image(--):  61.6  | Qwen-Image(--):  67.7  | **Bagel(+1)**:       64.5  | Qwen-Image(--):   66.8  |
> |4| Hidream(--):     53.3  | Hidream(--):     57.1  | **Flux(+1)**:        89.3  | **Hidream(-1)**:     77.8  | Hidream(--):     55.9  | Hidream(--):     64.6  | **Qwen-Image(-1)**:  63.6  | Bagel(--):        66.0  |
> |5| Bagel(--):       51.3  | Bagel(--):       52.4  | **Qwen-Image(-1)**:  88.3  | **SD3-Medium(+1)**:  70.2  | Bagel(--):       54.4  | Bagel(--):       63.9  | Hidream(--):     54.1  | Hidream(--):      59.1  |
> |6| Flux(--):        47.1  | Flux(--):        47.8  | **Bagel(+1)**:       82.2  | **Flux(-1)**:        68.9  | Flux(--):        47.0  | Flux(--):        58.2  | **Flux(+1)**:        53.2  | **Flux(+1)**:         58.4  |
> |7| SD3-Medium(--):  40.4  | SD3-Medium(--):  47.1  | **SD3-Medium(-1)**:  80.3  | Bagel(--):       51.9  | SD3-Medium(--):  44.0  | SD3-Medium(--):  51.2  | **SD3-Medium(-1)**:  51.0  | **SD3-Medium(-1)**:   56.6  |
> |8| Janus(--):       33.7  | Janus(--):       34.5  | Janus(--):       70.2  | Janus(--):       41.9  | Janus(--):       39.8  | Janus(--):       45.5  | Janus(--):       48.1  | Janus(--):        51.1  |
>
>
> **Table 3. Human Correlations using different MLLM evaluators**
> |MLLM| Idiom | Idiom | Textual | Textual | Entity | Entity | Scientific | Scientific | Average | Average |
> |------------|------------|------------|------------|------------|------------|------------|------------|------------|------------|------------|
> || τ  | ρ  |  τ  | ρ  | τ  | ρ  | τ  | ρ  | τ  | ρ | τ  | ρ |
> |Qwen2.5-VL|**0.5095**|**0.6792**|0.6051|**0.7686**|0.4767|0.6242|**0.4984**|**0.6438**|0.5224|**0.6790**
> |GPT-5.1   |0.4943 |0.6659|**0.6052** |0.7493|**0.5432**|**0.6831**|0.4704|0.6168|**0.5283**|0.6788
> |LLaVA-1V     |0.4664|0.6202|0.5561|0.6936|0.5116|0.6377|0.4342|0.5802|0.4921 |0.6329

---

> ### Author Response · Authors · 2025-11-23
> **Response to Reviewer jgck (3/5)**
>
> > **Q2**: Sensitivity analysis for score weights
>
> **A2**: We thank the reviewer for this valuable feedback regarding the weights of our T2I-ReasonScore.
>
> 1. **Rationale for Initial Weights**:
> Our motivation is to evaluate reasoning abilities of T2I generation. Therefore, we assigned a **high weight (0.7) to the core reasoning** component to ensure it is the dominant factor in the score. The secondary weight (0.2) for detail faithfulness accounts for other important contextual elements, and the minor weight (0.1) for image quality acknowledges its role in human preference without allowing it to overshadow semantic correctness.
>
> 2. **Sensitivity Analysis and Findings**:
> We conducted a comprehensive sensitivity analysis by testing multiple weight combinations for each dimension. **Tables 4.1 to 4.4** display the model rankings (including scores) and human correlations for each weight combination across the four dimensions. The key findings are summarized below:
>    - **Robustness of Rankings**: Across all dimensions, **the model rankings remain largely unchanged** under reasonable weight ranges (indicated by human correlation). Minor rank swaps (by 1 position) occur only between adjacent models with very close scores, and typically only when the reasoning weight is significantly reduced (e.g., to 0.3), which contradicts our motivation.
>    - **correlation with Human Judgment**: **The correlation with human scores remains stable across a range of weights.** However, we observed a consistent trend: reducing the weight for reasoning consistently leads to a decrease in human correlation and an increase in models' final scores. This empirically validates our design choice to prioritize reasoning and reinforces our paper's central finding that reasoning is the primary bottleneck for current models.
>    - **Dimension-Specific Results**:
>      - Idiom & Entity Reasoning: Model rankings are entirely unaffected by weight changes.
>      - Textual Image Design & Scientific Reasoning: Rankings show minimal sensitivity, with a single pairwise swap occurring only under extreme weighting that also yields the lowest human correlation.
>
> **Conclusion**: **The sensitivity analysis confirms that our overall model rankings are robust.** The chosen weights [0.9, 0.1] or [0.7, 0.2, 0.1] effectively capture the benchmark's focus on reasoning while maintaining high agreement with human judgment. Although the correlation at these weights may not be the highest among all combinations, we maintain them to avoid overfitting to the human scores from our specific evaluation set.

---

> ### Author Response · Authors · 2025-11-23
> **Response to Reviewer jgck (4/5)**
>
> **Table 4.1. Weight Sensitivity Analysis - Idiom**
> |Weights| [0.9, 0.1] - ours|   [0.7, 0.3]   | [0.5, 0.5] | [0.3, 0.7] |
> |------------|------------|------------|------------|------------|
> |**correlation (τ,ρ)**| 0.5095, 0.6792 | **0.5142, 0.6805**  |  0.4914, 0.6570 | 0.4425, 0.5958 |
> |No.1| Nano-Banana: 80.4 |  Nano-Banana(--): 83.6 |  Nano-Banana(--): 86.7 |  Nano-Banana(--): 89.9 |
> |No.2| GPT-Image-1: 77.6 |  GPT-Image-1(--): 81.3 |  GPT-Image-1(--): 85.1 |  GPT-Image-1(--): 88.9 |
> |No.3| Qwen-Image:  55.5 |  Qwen-Image(--):  63.1 |  Qwen-Image(--):  70.7 |  Qwen-Image(--):  78.3 |
> |No.4| Hidream:     52.4 |  Hidream(--):     60.1 |  Hidream(--):     67.9 |  Hidream(--):     75.6 |
> |No.5| Bagel:       48.6 |  Bagel(--):       56.5 |  Bagel(--):       64.4 |  Bagel(--):       72.5 |
> |No.6| Flux:        45.1 |  Flux(--):        53.6 |  Flux(--):        62.0 |  Flux(--):        70.4 |
> |No.7| SD3-Medium:  40.5 |  SD3-Medium(--):  49.6 |  SD3-Medium(--):  58.7 |  SD3-Medium(--):  67.8 |
> |No.8| Janus:       30.7 |  Janus(--):       41.2 |  Janus(--):       51.8 |  Janus(--):       62.3 |
>
>
> **Table 4.2. Weight Sensitivity Analysis - Textual Image Design**
> |Weight | [0.9, 0.1] - ours| [0.7, 0.3] | [0.5, 0.5] | [0.3, 0.7] |
> |------------|------------|------------|------------|------------|
> |**correlation (τ,ρ)**| 0.6051, 0.7686 | **0.6175, 0.7806**  |  0.6097, 0.7754 | 0.5720, 0.7422 |
> |No.1| Nano-Banana: 93.4 |  Nano-Banana(--): 94.0 |  Nano-Banana(--): 94.5 |  Nano-Banana(--):  95.1  |
> |No.2| GPT-Image-1: 88.0 |  GPT-Image-1(--): 90.1 |  GPT-Image-1(--): 92.3 |  GPT-Image-1(--):  94.4  |
> |No.3| Hidream:     73.6 |  Hidream(--):     76.2 |  Hidream(--):     78.9 |  Hidream(--):      81.5  |
> |No.4| Qwen-Image:  72.9 |  Qwen-Image(--):  74.8 |  Qwen-Image(--):  76.6 |  Qwen-Image(--):   78.5  |
> |No.5| Flux:        66.1 |  Flux(--):        68.0 |  Flux(--):        69.8 |  **SD3-Medium(+1)**:   71.7  |
> |No.6| SD3-Medium:  61.9 |  SD3-Medium(--):  64.0 |  SD3-Medium(--):  66.1 |  **Flux(-1)**:         68.2  |
> |No.7| Bagel:       46.9 |  Bagel(--):       52.9 |  Bagel(--):       58.8 |  Bagel(--):        64.8  |
> |No.8| Janus:       40.5 |  Janus(--):       47.3 |  Janus(--):       54.0 |  Janus(--):        60.8  |
>
>
> **Table 4.3. Weight Sensitivity Analysis - Entity**
> |Weights| [0.7, 0.2, 0.1] - ours| [0.45, 0.45, 0.1] | [0.2, 0.7, 0.1] | [0.35, 0.35, 0.3] | [0.2, 0.1, 0.7] |
> |------------|------------|------------|------------|------------|------------|
> |**correlation(τ,ρ)**| 0.4767, 0.6242 | 0.4828, 0.6314  |  0.4711, 0.6198  | **0.4875, 0.6363** | 0.4502, 0.5907 |
> |No.1| Nano-Banana: 83.6  | Nano-Banana(--): 85.7 | Nano-Banana(--): 87.9 | Nano-Banana(--): 88.5 | Nano-Banana(--): 93.6 |
> |No.2| GPT-Image-1: 79.1  | GPT-Image-1(--): 80.1 | GPT-Image-1(--): 81.1 | GPT-Image-1(--): 83.8 | GPT-Image-1(--): 90.9 |
> |No.3| Qwen-Image:  62.6  | Qwen-Image(--):  68.2 | Qwen-Image(--):  73.8 | Qwen-Image(--):  74.4 | Qwen-Image(--):  85.7 |
> |No.4| Hidream:     56.5  | Hidream(--):     62.3 | Hidream(--):     68.1 | Hidream(--):     69.3 | Hidream(--):     82.3 |
> |No.5| Bagel:       55.3  | Bagel(--):       59.1 | Bagel(--):       62.8 | Bagel(--):       66.3 | Bagel(--):       80.0 |
> |No.6| Flux:        47.8  | Flux(--):        53.9 | Flux(--):        60.1 | Flux(--):        62.3 | Flux(--):        77.7 |
> |No.7| SD3-Medium:  45.8  | SD3-Medium(--):  50.7 | SD3-Medium(--):  55.7 | SD3-Medium(--):  59.5 | SD3-Medium(--):  76.0 |
> |No.8| Janus:       41.7  | Janus(--):       47.7 | Janus(--):       53.6 | Janus(--):       56.5 | Janus(--):       73.1 |
>
>
> **Table 4.4. Weight Sensitivity Analysis - Scientific**
> |Weights| [0.7, 0.2, 0.1] - ours| [0.45, 0.45, 0.1] | [0.2, 0.7, 0.1] | [0.35, 0.35, 0.3] | [0.2, 0.1, 0.7] |
> |------------|------------|------------|------------|------------|------------|
> |**correlation(τ,ρ)**|**0.4984, 0.6438** | 0.4807, 0.6208  |  0.3985, 0.5283  | 0.4802, 0.6290 | 0.4220, 0.5681 |
> |No.1| Nano-Banana: 78.6 | Nano-Banana(--): 82.7 | Nano-Banana(--): 86.8 | Nano-Banana(--): 85.4 | Nano-Banana(--): 89.9 |
> |No.2| GPT-Image-1: 75.3 | GPT-Image-1(--): 80.2 | GPT-Image-1(--): 85.1 | GPT-Image-1(--): 83.3 | GPT-Image-1(--): 88.6 |
> |No.3| Qwen-Image:  61.6 | Qwen-Image(--):  69.0 | Qwen-Image(--):  76.5 | Qwen-Image(--):  73.1 | Qwen-Image(--):  79.8 |
> |No.4| Bagel:       59.5 | Bagel(--):       64.2 | Bagel(--):       68.9 | Bagel(--):       69.5 | Bagel(--):       79.3 |
> |No.5| Hidream:     54.5 | Hidream(--):     61.3 | Hidream(--):     68.2 | Hidream(--):     66.5 | Hidream(--):     75.4 |
> |No.6| SD3-Medium:  52.0 | **Flux(+1)**:        59.7 | **Flux(+1)**:        67.7 | **Flux(+1)**:        64.9 | **Flux(+1)**:        73.7 |
> |No.7| Flux:        51.7 | **SD3-Medium(-1)**:  59.2 | **SD3-Medium(-1)**:  66.3 | **SD3-Medium(-1)**:  64.2 | **SD3-Medium(-1)**:  72.7 |
> |No.8| Janus:       46.4 | Janus(--):       52.9 | Janus(--):       59.5 | Janus(--):       58.5 | Janus(--):       68.2 |

---

> ### Author Response · Authors · 2025-11-23
> **Response to Reviewer jgck (5/5)**
>
> > **Q3**:  Oversimplified interpretation of results
>
> **A3**: We sincerely thank the reviewer for this insightful comment. We totally agree with the reviewer's opinion. The performance change is more plausibly explained by a mismatch between the format of the verbose, rewritten prompts and the models' training data, which can constrain their inherent reasoning capabilities. We have revised the main text to reflect this interpretation.
>
> > **Q4**: Reasoning vs. memorization.
>
> **A4**: We thank the reviewer for this critical question. We think that it is almost impossible to disentangle reasoning from memorization completely. T2I-ReasonBench aims to create scenarios that require models to integrate domain knowledge, infer implicit meaning, and resolve contextual ambiguities. In this process, knowledge serves as a necessary prerequisite for reasoning, as stated in our abstract, introduction and Section 3.1 problem definition. The tasks are constructed such that successful performance is unlikely through memorization alone, thereby pushing models beyond simple recall.
>
> 1. **Idiom Task Design that Penalizes Memorization**:
> The Idiom Interpretation dimension is explicitly designed to penalize simple associative recall and test for genuine reasoning.
> An idiom's literal meaning (e.g., "a piece of cake," "the cat out of the bag") is highly likely to exist in training data, making it easy for models to generate.
> In contrast, its metaphorical meaning is less likely existing in the text-image dataset, possibly the corpus data for UMM pretraining.
> Consequently, a model must perform a two-step reasoning process: first, it must select the correct meaning based on the context, and second, it must plan the visual content and avoid the literal depiction. Both steps require reasoning beyond mere memorization. This is also addressed in our evaluation, for example:
>    - the prompt *"I volunteered for too many projects and quickly realized I'd bitten off more than I could chew."* shold convey a sense of being overwhelmed by excessive responsibilities, one of its question-criteron pairs is:
> *"Does the composition avoid a literal interpretation of 'biting' or 'chewing' while maintaining the idiom’s core meaning (excessive workload)?",
> "score 1: No mouth/food literalism; focus on workload. score 0.5: Minor irrelevant elements. 0: Depicts eating/biting."*
>    - Another example is prompt *"Despite our careful planning, John let the cat out of the bag during dinner last night".* It should depict John revealing a secret during a dinner gathering, causing visible reactions of surprise/shock among others present. one of its question-criteron pairs is:
> *"Is there any literal depiction of 'letting a cat out of a bag'?", "Score 1: No cat/bag present. Score 0.5: Only one element (cat or bag) unrelated to revelation. Score 0: Literal cat emering from bag shown".*
>
> 2. **Reasoning in Entity dimension**:
> Similarly, the Entity-Reasoning dimension requires connecting multiple facts, not just recalling a single entity. For example:
>    - The prompt *"The basketball team winning 6 NBA championships in the 1990s"* requires a model to first recall championship data for a decade (1990s) and then identify the unique team fulfilling that condition (6 times). This is a multi-step query.
>    - We observe that even models with knowledge can fail, indicating a reasoning breakdown. For instance given the prompt *"The Mexican fruit with leathery skin and blood-red pulp containing edible black seeds"* to GPT-Image-1, whose text answer "red dragon fruit" is correct, but still generate a fruit with some correct features but definitely not red dragon fruit, demonstrating a failure in knowledge integration.

---

### Official Review · Reviewer_CuYc · 2025-11-01

**Soundness:** 3
**Presentation:** 4
**Contribution:** 3
**Rating:** 6
**Confidence:** 4

**Summary:**

This paper introduces T2I-ReasonBench, a new benchmark to evaluate reasoning capabilities in text-to-image generation across four dimensions: Idiom Interpretation, Textual Image Design, Entity-Reasoning, and Scientific-Reasoning. It proposes a two-stage evaluation pipeline where a large language model (LLM) first generates prompt-specific question–criterion pairs, and then a multimodal LLM scores the generated image against these criteria, yielding a quantitative metric called T2I-ReasonScore. Using this framework, the authors evaluate 16 state-of-the-art T2I models and find that existing models struggle with prompts requiring deeper reasoning. The results suggest that the primary bottleneck of current T2I models lies in their reasoning ability rather than low-level image generation quality.

**Strengths:**

1. This work addresses a critical gap by focusing on reasoning capabilities in T2I generation, an aspect that previous benchmarks largely ignored. By challenging models with idioms, complex design tasks, entity knowledge, and scientific scenarios, it goes beyond surface-level prompt-to-image alignment to evaluate deeper understanding and inference in image generation. The benchmark comprises 800 carefully curated prompts spanning four diverse reasoning dimensions. This thorough coverage ensures that a wide range of reasoning skills (from interpreting figurative language to applying scientific laws) are tested, providing a balanced and systematic assessment of models under consistent conditions.
2. The paper introduces a two-stage LLM-based evaluation pipeline that produces a fine-grained metric tailored to reasoning-intensive tasks. Notably, the authors validate this automatic metric by showing it correlates more strongly with human judgment than standard metrics like CLIP or VQA scores, lending credibility to their approach.
3. The authors evaluate 16 modern T2I and multimodal models, providing a rich comparative analysis of their reasoning performance. The experiments yield valuable insights – for example, even advanced models are shown to be limited by reasoning rather than generative fidelity – which can guide future research.

**Weaknesses:**

1. The evaluation framework heavily relies on an LLM and a multimodal model as judges, which introduces potential bias and uncertainty in the scoring. Although the authors demonstrate that their metric aligns well with human evaluations, the dependence on AI evaluators (which have their own limitations) raises concerns about whether the scores always faithfully reflect human-perceived reasoning quality. The two-stage evaluation process is fairly complex and computationally intensive. It depends on a specific LLM (DeepSeek-R1) and a large vision-language model (Qwen2.5-VL) for each evaluation, which may hinder reproducibility and adoption – researchers without access to these models or similar compute resources could find it difficult to apply the benchmark or reproduce the exact scoring.
2. The chosen four dimensions, while sensible, may not cover the full spectrum of reasoning needed for image generation. For instance, certain forms of commonsense or social reasoning might fall outside these categories, suggesting that T2I-ReasonBench could be further expanded to ensure no important reasoning skill is left untested.

**Questions:**

1. How do the authors ensure that each prompt cleanly fits into one of the four reasoning categories without overlap? For example, if an idiom prompt also involves some scientific or commonsense reasoning, was it categorized in a single dimension, and what criteria were used to handle such overlaps or ambiguities in prompt classification?

---

> ### Author Response · Authors · 2025-11-27
> **Response to Reviewer CuYc (1/2)**
>
> > **Q1**: Verify reliability of evaluation metric and update human correlation as new models become available.
>
> **A1**:
> We thank the reviewer for raising the concerns regarding potential evaluator bias and computational demands in our framework. To systematically address these issues, we conducted extensive validation experiments.
>
> 1. **Expand human evaluation**:
> We first expanded our human evaluation set by incorporating 3 additional models, resulting in a total of 8 models, then conducted experiments using 2 different LLMs (DeepSeek-R1 and the recently released GPT-5.1) and 3 different MLLMs (Qwen2.5-VL, GPT-5.1, and LLaVA-OneVision [A]), resulting in 6 unique evaluation pipeline configurations. The human correlations for all 6 pipelines are presented in Table 1.
>
>    Our analysis reveals two key findings:
>
>    -  Regarding the Stage 1 LLM: **GPT-5.1 generally outperforms DeepSeek-R1**. When using the same MLLM evaluator, pipelines with question-criterion sets generated by GPT-5.1 achieve higher correlations in almost all dimensions.
>
>    -  Regarding the Stage 2 MLLM: **The open-source model Qwen2.5-VL demonstrates performance comparable to the commercial GPT-5.1**. When provided with questions from GPT-5.1, Qwen2.5-VL's correlation is even higher.
>
>
> 2. **Manually check question-criterion samples generated by different LLMs**:
> To further investigate the quality of the generated questions & criteria, we conducted a human evaluation on 80 samples from each LLM. We found that **both LLMs produced reasonable, non-hallucinated questions and criteria for all prompts**, with no reasoning errors or knowledge gaps. The questions generated by both models were largely overlapping and addressed the same core aspects of the prompts. However, GPT-5.1 demonstrated superior quality in two ways:
>
>    - Its **questions and criteria were more detailed and specific**, with clearer descriptions of expected visual elements.
>
>    - The **structure was more refined**, with each question being more "atomic" (covering the smallest semantic unit) and exhibiting less overlap between questions within a set.
>
>    For example, for the idiom prompt "After three weeks of burning the midnight oil, she finally submitted her dissertation":
>
>    * GPT-5.1's first two questions are:
>
>       * "q1": "Does the image clearly show a female main character as the central figure?"
>       * "q2": "Is it clear that she is working on or has just completed an academic dissertation or similar substantial written work?"
>
>    * DeepSeek-R1's first two questions are:
>       * "q1": "Does the image clearly show a woman engaged in academic work?" **(combines 'woman' and 'work')**
>       * "q2": "Are academic materials (books/papers/laptop) prominently featured?" **(overlaps with the academic theme of q1)**
>
>
> While GPT-5.1 delivers better performance, it was not available at the time of our initial experiments. A key reason for our original choice of **DeepSeek-R1 and Qwen2.5-VL is that they are open-source models**, more accessible for the research community.
>
> We fully acknowledge that the AI field evolves rapidly. Therefore, in the final version, we commit to releasing all human scores and will periodically update the correlation rankings with new LLM and MLLM combinations. This will ensure our benchmark remains a credible, transparent, and up-to-date resource, **allowing researchers to select evaluation models based on their computational resources and budgets**.
>
>
> **Table 1. Human Correlations using different LLM and MLLM**
> |LLM|MLLM| Idiom | Idiom | Textual | Textual | Entity | Entity | Scientific | Scientific | Average | Average |
> |------------|------------|------------|------------|------------|------------|------------|------------|------------|------------|------------|------------|
> | || τ  | ρ  |  τ  | ρ  | τ  | ρ  | τ  | ρ  | τ  | ρ | τ  | ρ |
> |DeepSeek|Qwen2.5-VL|0.5095|0.6792|0.6051|0.7686|0.4767|0.6242|0.4984|0.6438|0.5224 |**0.6790**
> |DeepSeek|GPT-5.1   |0.4943|0.6659|0.6052 |0.7493|0.5432|0.6831|0.4704|0.6168|**0.5283** |0.6788
> |DeepSeek|LLaVA-1V |0.4664|0.6202|0.5561|0.6936|0.5116|0.6377|0.4342|0.5802|0.4921 |0.6329
> |GPT-5.1 |Qwen2.5-VL|0.5391|0.7004|0.6372|0.8002|0.5937|0.7431|0.5062|0.6592|**0.5691** |**0.7257**
> |GPT-5.1 |GPT-5.1   |0.5576|0.7168|0.6446|0.8076|0.6077|0.7642|0.4589|0.6098|0.5672|0.7246
> |GPT-5.1 |LLaVA-1V |0.5165|0.6594|0.5343|0.6619|0.5762|0.7074|0.3952|0.5130|0.5056 |0.6354
>
> -----------------------
> [A] Li, Bo, et al. Llava-onevision: Easy visual task transfer. 2024

---

> ### Author Response · Authors · 2025-11-27
> **Response to Reviewer CuYc (2/2)**
>
> > **Q2**: Prompt Categorization and Potential Overlaps
>
> **A2**:
> To clarify, the prompts for each dimension are carefully and specifically designed, with **each dimension having its own distinct source**. For example, idiom prompts are taken from a comprehensive book of idioms. For scientific and entity reasoning, prompts are first divided into subcategories (such as physics, chemistry, or plant, animal). Then, human experts create the initial prompts, which are further expanded by LLMs, as detailed in Section 3.2 and Appendix A. Because of this structured process, there should be no overlap in prompts classification.
>
>  We appreciate the reviewer’s suggestion to expand the benchmark by incorporating more reasoning skills. In future work, we plan to add more dimensions and include a range of difficulty levels.
>
> Please don't hesitate to let us know if you have further questions. Thanks again for reviewing this paper!

---

### Official Review · Reviewer_vSRa · 2025-11-01

**Soundness:** 2
**Presentation:** 2
**Contribution:** 2
**Rating:** 2
**Confidence:** 5

**Summary:**

This paper introduces T2I-ReasonBench, a new benchmark aimed at evaluating the reasoning capabilities of text-to-image (T2I) generative models. Existing benchmarks primarily assess literal text-image alignment (e.g., object color, count, or position). The authors argue that true understanding requires reasoning beyond surface text cues.

T2I-ReasonBench covers four reasoning dimensions:
1. Idiom Interpretation — understanding figurative language and implicit meaning.
2. extual Image Design — reasoning about communicative intent and integrating text visually.
3. Entity-Reasoning — inferring unstated entities using world knowledge.
4. Scientific-Reasoning — applying physical or scientific principles.

The benchmark includes 800 prompts and introduces a two-stage evaluation framework:
Stage 1: An LLM generates question-criterion pairs per prompt to guide evaluation.
Stage 2: An MLLM scores generated images against those criteria, producing the composite metric T2I-ReasonScore (weighted by reasoning, details, and quality).

**Strengths:**

- It moves beyond compositionality and literal alignment toward reasoning-aware evaluation — an underexplored but crucial capability for T2I systems.
- It covers a wide range of models, including diffusion, unified, and proprietary systems.

**Weaknesses:**

- The work relies excessively on LLMs, yet lacks a thorough verification process for hallucinations or incorrect responses.
- Although several other related metrics already exist—such as TIFA [1], and I-HallA [2]—the paper does not provide any comparison with them.
- The paper attempts to tackle four major challenges at once, resulting in an unfocused contribution. Since the benchmark is not large-scale, it should have been carefully curated; however, it ends up being of ambiguous size and quality.
- For the Scientific Reasoning component, experts should have been consulted or a rigorous human verification process should have been conducted. For example, “A trampoline with an iron ball on it” only causes the surface to deform under Earth-like gravity; in an environment with different gravity, the outcome would differ. Scientific reasoning requires precise conditions and verification.
- Figure 2 is supposed to illustrate the overall framework and methodology of the paper, yet it only presents a high-level concept of the data collection process without concrete details. The caption also lacks substance and clarity.
- I don't think the correlation score results are high enough.

[1] Hu, Yushi, et al. "Tifa: Accurate and interpretable text-to-image faithfulness evaluation with question answering." Proceedings of the IEEE/CVF International Conference on Computer Vision. 2023.

[2] Lim, Youngsun, Hojun Choi, and Hyunjung Shim. "Evaluating Image Hallucination in Text-to-Image Generation with Question-Answering." Proceedings of the AAAI Conference on Artificial Intelligence. Vol. 39. No. 25. 2025.

**Questions:**

- As far as I know, even LLMs have not yet mastered idiom interpretation. How can a T2I model be expected to capture such figurative meaning accurately? Moreover, I don’t understand the rationale for translating those idiomatic expressions into images in the first place.
- In the Textual Image Design section, how would one evaluate a prompt like “Create a minimalist promotional poster for a workshop on simplicity in design”? This is a topic on which even humans lack consensus, yet the paper relies solely on a rudimentary LLM-based metric. Do the authors genuinely believe this provides a convincing or meaningful evaluation?

---

> ### Author Response · Authors · 2025-11-27
> **Response to Reviewer vSRa (1/3)**
>
> > **Q1**: Expand human evaluation and verify LLM reliability
>
> **A1**:
> We thank the reviewer for raising this concern regarding potential LLM hallucinations and the need for thorough verification.
>
> To rigorously validate the reliability of our evaluation framework, we implemented two steps:
>
>
> - **Step 1: Expand human evaluation and analyze correlations**:
>
>    We first expanded our human evaluation set by incorporating 3 additional models, resulting in a total of 8 models, then conducted experiments using 2 different LLMs (DeepSeek-R1 and the recently released GPT-5.1) and 3 different MLLMs (Qwen2.5-VL, GPT-5.1, and LLaVA-OneVision [A]), resulting in 6 unique evaluation pipeline configurations. The human correlations for all 6 pipelines are presented in **Table 1**.
>
>    Our analysis reveals two key findings:
>
>    1. Regarding the Stage 1 LLM: **GPT-5.1 generally outperforms DeepSeek-R1**. When using the same MLLM evaluator, pipelines with question-criterion sets generated by GPT-5.1 achieve higher correlations in almost all dimensions.
>
>    2. Regarding the Stage 2 MLLM: **The open-source model Qwen2.5-VL demonstrates performance comparable to the commercial GPT-5.1**. When provided with questions from GPT-5.1, Qwen2.5-VL's correlation is even higher.
>
> - **Step 2: Manually check and compare question-criterion samples generated by different LLMs**:
>
>    To further investigate the quality of the generated questions & criteria, we conducted a human evaluation on 80 samples from each LLM. We found that both **LLMs produced reasonable, non-hallucinated questions and criteria for all prompts**, with no instances of reasoning errors or critical knowledge gaps. The questions generated by both models were largely overlapping and addressed the same core aspects of the prompts.
>
>    However, GPT-5.1 demonstrated better quality in two ways:
>
>    1. Its **questions and criteria are more detailed and specific**, with clearer descriptions of expected visual elements.
>
>    2. The **structure is more refined**, with each question being more "atomic" (covering the smallest semantic unit) and exhibiting less overlap between questions within a set.
>
>       For example, for the idiom prompt *"After three weeks of burning the midnight oil, she finally submitted her dissertation"*:
>
>       * GPT-5.1's first two questions are:
>
>          * "q1": "Does the image clearly show a female main character as the central figure?"
>          * "q2": "Is it clear that she is working on or has just completed an academic dissertation or similar substantial written work?"
>
>       * DeepSeek-R1's first two questions are:
>          * "q1": "Does the image clearly show a woman engaged in academic work?" **(combines 'woman' and 'work')**
>          * "q2": "Are academic materials (books/papers/laptop) prominently featured?" **(overlaps with the academic theme of q1)**
>
>
> While GPT-5.1 delivers better performance as stage 1 LLM, it was not available at the time of our initial experiments. A key reason for our original choice of DeepSeek-R1 and Qwen2.5-VL is that they are open-source models, more accessible for the research community.
>
> We fully acknowledge that the AI field evolves rapidly. Therefore, in the final version, we commit to **releasing all human scores and will periodically update the correlation rankings with new LLM and MLLM combinations**. This will ensure our benchmark remains a credible, transparent, and up-to-date resource, allowing researchers to select evaluation models based on their computational resources and budgets.
>
> **Table 1. Human Correlations using different LLMs and MLLMs**
> |LLM|MLLM| Idiom | Idiom | Textual | Textual | Entity | Entity | Scientific | Scientific | Average | Average |
> |------------|------------|------------|------------|------------|------------|------------|------------|------------|------------|------------|------------|
> | || τ  | ρ  |  τ  | ρ  | τ  | ρ  | τ  | ρ  | τ  | ρ | τ  | ρ |
> |DeepSeek|Qwen2.5-VL|0.5095|0.6792|0.6051|0.7686|0.4767|0.6242|0.4984|0.6438|0.5224 |**0.6790**
> |DeepSeek|GPT-5.1   |0.4943|0.6659|0.6052 |0.7493|0.5432|0.6831|0.4704|0.6168|**0.5283** |0.6788
> |DeepSeek|LLaVA-1V |0.4664|0.6202|0.5561|0.6936|0.5116|0.6377|0.4342|0.5802|0.4921 |0.6329
> |GPT-5.1 |Qwen2.5-VL|0.5391|0.7004|0.6372|0.8002|0.5937|0.7431|0.5062|0.6592|**0.5691** |**0.7257**
> |GPT-5.1 |GPT-5.1   |0.5576|0.7168|0.6446|0.8076|0.6077|0.7642|0.4589|0.6098|0.5672|0.7246
> |GPT-5.1 |LLaVA-1V |0.5165|0.6594|0.5343|0.6619|0.5762|0.7074|0.3952|0.5130|0.5056 |0.6354
>
> ---------------------------------------------------------------
> [A] Li, Bo, et al. Llava-onevision: Easy visual task transfer. 2024

---

> ### Author Response · Authors · 2025-11-27
> **Response to Reviewer vSRa (2/3)**
>
> > **Q2**: Questions on Benchmark Design
>
> **A2**:
> We thank the reviewer for questions regarding the design of our four reasoning dimensions. Below, we address each point in detail.
>
> 1. **Conditions for Scientific Reasoning:**
>
>      - We agree that scientific reasoning requires pre-defined conditions. However, **an infinite number of conditions can theoretically be specified for any given scenario**. For example, the prompt "A trampoline with an iron ball on it" could have its outcome altered by specifying gravity, relative sizes, weights or materials. Therefore, **in the absence of explicit conditions, it is natural to assume a default real-world setting (Earth gravity, common sizes) where models should perform the most probable reasoning trace**. This dimension evaluates a model's ability to apply fundamental scientific principles, not to exhaustively account for every possible conditions.
>
>
> 2. **Design and Rationale for Idiom Interpretation:**
>
>    - Design Process: The idioms we manually selected are all commonly used in daily life. When creating each prompt, the idiom and its actual meaning were given to the LLM, and we manually checked all prompts to ensure they provide contextual clues without directly revealing the idiom's meaning. **At evaluation, the LLM receives the prompt, the specific idiom, and its actual meaning to ensure accurate question-criterion generation** (detailed in main text Appendix A and Table 4).
>
>    - Capability Verification: We conducted human verification on 80 samples. The evaluation questions & criteria are all reasonable without hallucination. This confirms that **LLMs can reliably interpret idioms**.
>
>    - Research Motivation: Idioms are pervasive in everyday language. While expert users and researchers may understand the limitations of current T2I tools, the majority of general users may intuitively input prompts containing idiomatic expressions that fall outside the training set of current models. **As LLMs are required to be more socialized [A] and emotional intelligent [B, C], why we cannot expect T2I models to capture figurative meaning?** It is difficult to visualize abstract metaphorical concepts, but this shouldn't be the reason precluding us from exploring this ability for T2I models. Therefore, we believe this dimension meaningfully expands the frontier of what we expect from reasoning-informed T2I generation.
>
> 3. **Evaluation of Textual Image Design:**
>    - We want to firstly clarify that all prompts in this dimension derive from real-world images with specific communicative functions (Appendix A).
>
>    - **Rather than prescribing visual content, the core of evaluation is to assess whether generated images fulfill their intended communicative purpose**. The evaluation questions and criteria focus on functional elements. They allows complete creation freedom to designers. For the workshop poster example, the LLM-generated criteria evaluate:
>       - Clear presentation of workshop topic
>       - Explicit identification as an event
>       - Inclusion of essential details (date/time/location)
>       - Consistency with minimalist design principles
>
>       It can be seen the **evaluation is grounded in broadly accepted design requirements for the prompt.**
>
> 4. **Focus and Scale of the benchmark**:
>    - We respectfully disagree that tackling four reasoning dimensions leads to an unfocused contribution. Rather, these dimensions are carefully chosen to provide a holistic evaluation of different facets of reasoning, from linguistic and creative design to factual and scientific reasoning. They push T2I generation a step forward from literal prompt-following to reasoning implicit meaning.
>
>    - Each of the 800 prompts was meticulously designed and manually verified to ensure it genuinely tests reasoning capabilities. The size is also consistent with other reasoning benchmarks [D-F] (700-1000 prompts) in this field.
>
> ---------------------------------------------------------------
> [A] Bisk, Yonatan, et al. Experience grounds language. 2020
>
> [B] Sabour, Sahand, et al. Emobench: Evaluating the emotional intelligence of large language models. 2024
>
> [C] Chen, Zhuang, et al. Tombench: Benchmarking theory of mind in large language models. 2024
>
> [D] Meng, Fanqing, et al. PhyBench: A physical commonsense benchmark for evaluating text-to-image models." 2024
>
> [E] Niu, Yuwei, et al. WiSE: A world knowledge-informed semantic evaluation for text-to-image generation." 2025
>
> [F] Fu, Xingyu, et al. Commonsense-T2I challenge: Can text-to-image generation models understand commonsense?. 2024

---

> ### Author Response · Authors · 2025-11-27
> **Response to Reviewer vSRa (3/3)**
>
> > **Q3**: Clarification on Framework Documentation
>
> **A3**:
> The overall framework is documented across following sections: Section 3.1 establishes the benchmark's motivation, Section 3.2 details the construction steps including all tools used, and Appendix A provides details about the data collection process.
> The evaluation methodology is presented in Section 4.
> We thank the reviewer for feedback regarding Figure 2 and will revise it to enhance clarity.
>
> > **Q4**: Comparison with Existing Metrics
>
> **A4**: Thank the reviewer for raising this point. We will add a subsection in Related Work comparing with other established metrics that also employ a two-stage evaluation approach.

---

### Author Response · Authors · 2025-12-03
**summary of rebuttals**

We would like to thank all the reviewers for your thoughtful, detailed, and constructive feedback on our paper.
We have carefully considered all the comments and have conducted significant additional experiments and analyses to address the key concerns raised. Below, we summarize the main recurring questions and the substantial revisions we have made in response:

1. **Concern: Reliability of the AI-Evaluating-AI Evaluation Framework**
- Reviewers’ Point: The two-stage evaluation pipeline (LLM-generated criteria + MLLM scoring) could be biased or suffer from compounded errors.
- Our Response: We conducted extensive validation:
   - **Expanded human evaluation from 5 to 8 models, add human evaluation details in Appendix D**.
   - **Conducted human correlation analysis on 12 different LLM-MLLM combinations, including additional models like: Qwen3-VL, GPT-5.1, Gemini-2.5-pro, and LLaVA-OneVision, finally selected an optimal LLM-MLLM combination for each dimension and revised the scores in Table 2, Table 3 of main text (Details in the added Appendix B: Selection of Models for Evaluation).**
   - Model rankings are robust across different MLLM evaluators and Qwen2.5-VL shows no measurable bias toward Qwen-Image.
   - Manually verified 80 question-criterion pairs per LLM and found them to be reasonable, specific, and free of hallucinations.
   - Commitment: We will release all human scores and periodically update correlation tables as new models emerge.

2. **Concern: Arbitrariness of Score Weights**
- Reviewers’ Point: The weights in T2I-ReasonScore (e.g., 0.7 for reasoning) were not justified with sensitivity analysis.
- Our Response:
   - **Conducted a full sensitivity analysis across multiple weight combinations.**
   - **Showed that model rankings are robust under reasonable weight ranges.**
   - Found that reducing the reasoning weight consistently lowers correlation with human judgment, validating our emphasis on reasoning as the primary bottleneck.


3. **Concern: Distinguishing Reasoning from Factual Recall:**
- Reviewers’ Point: Some tasks (e.g., Entity-Reasoning) might test memorized knowledge rather than genuine reasoning.
- Our Response:
   - Clarified that knowledge is a prerequisite, but reasoning is defined as the integration and application of knowledge to infer implicit meaning and generate a semantically faithful image.
   - Highlighted how our evaluation criteria actively penalize simple recall (e.g., idiom tasks require avoiding literal depictions)
   - Provided examples in Entity-Reasoning that go beyond single-fact retrieval.

4. **Concern: Interpretation of Results**
- Reviewers’ Point: The claim about Nano-Banana’s “superior internal reasoning” was overstated; performance changes may stem from prompt format mismatches.
- Our Response:
   - **Revised the text (Section 5.3) to present this interpretation**

5. Additional Revisions
   - We have **revised Section 4** to better highlight the metric definition.
   - We have **added a subsection in Related Work** comparing our framework with existing metrics (TIFA, I-HallA, etc.).

---

### Note · Authors · 2026-01-06

I have read and agree with the venue's withdrawal policy on behalf of myself and my co-authors.